# A CLIP-Powered Framework for Robust and Generalizable Data Selection

**Suorong Yang[1,2*], Peng Ye[2,3], Wanli Ouyang[2], Dongzhan Zhou[2†], Furao Shen[1†]**
[1] National Key Laboratory for Novel Software Technology, Nanjing University
[2] Shanghai Artificial Intelligence Laboratory
[3] The Chinese University of Hong Kong

## Abstract

Large-scale datasets have been pivotal to the advancements of deep learning models in recent years, but training on such large datasets inevitably incurs substantial storage and computational overhead. Meanwhile, real-world datasets often contain redundant and noisy data, imposing a negative impact on training efficiency and model performance. Data selection has shown promise in identifying the most representative samples from the entire dataset, which aims to minimize the performance gap with reduced training costs. Existing works typically rely on single-modality information to assign importance scores for individual samples, which may lead to inaccurate assessments, especially when dealing with noisy or corrupted samples. To address this limitation, we propose a novel CLIP-powered data selection framework that leverages multimodal information for more robust and generalizable sample selection. Specifically, our framework consists of three key modules—dataset adaptation, sample scoring, and selection optimization—that together harness extensive pre-trained multimodal knowledge to comprehensively assess sample influence and optimize the selection results through multi-objective optimization. Extensive experiments demonstrate that our approach consistently outperforms existing state-of-the-art baselines on various benchmark datasets. Notably, our method effectively removes noisy or damaged samples from the dataset, enabling it to achieve even higher performance with less data. This indicates that it is not only a way to accelerate training but can also improve overall data quality. The implementation is available at `https://github.com/Jackbrocp/clip-powered-data-selection`.

## 1 Introduction

Recent advancements in deep learning have been propelled by increasingly large and complex models that utilize vast datasets to achieve start-of-the-art performance Liu et al. (2024b); Touvron et al. (2023). However, this success normally comes with considerable costs for data storage and computational resources, which may even limit the deployment of models to specialized infrastructure and hinder their scalability across different applications. Moreover, real-world datasets often contain redundancy and noise, which can degrade the training efficiency and performance.

To alleviate the data redundancy issue and improve the training efficiency, there are typically two kinds of methods, i.e., dynamic pruning and data selection. Dynamic pruning methods Raju et al. (2021); Qin et al. (2024) aim to reduce training costs by dynamically selecting only the most influential samples from the dataset during training. Despite their effectiveness in accelerating training, they still face the cost of large-scale data storage, and their selected samples often lack the ability to generalize well across different training processes and architectures. In contrast, data selection methods Paul et al. (2021); Yang et al. (2023b); Tan et al. (2024); Xia et al. (2023) pre-select a fixed subset of the essential data points before the training begins, allowing the model to achieve performance comparable to that obtained with the full dataset. By focusing on the most critical data points, these methods ensure better generalization ability across various training scenarios.

---

[*] This work was done during his internship at Shanghai Artificial Intelligence Laboratory.
[†] Corresponding authors.

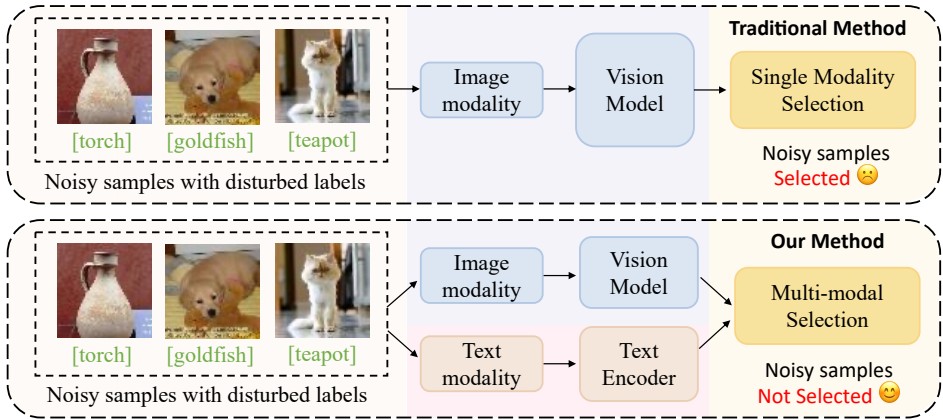

Figure 1: Benefit of multimodal selection. Traditional single-modality approaches (upper part) may struggle with noisy and corrupted data, while our multimodal method (lower part) identifies class-representative samples with diversity while effectively filtering out noise and corrupted data.

Existing data selection methods typically employ carefully designed criteria via three perspectives: importance scores Paul et al. (2021); Tan et al. (2024), image data distribution Zheng et al. (2023); Xia et al. (2023), and optimization-based functions Killamsetty et al. (2021b); Yang et al. (2023b). Although achieving promising results, these methods exhibit certain limitations. On one hand, relying solely on single-modality image information can lead to ambiguities, especially when noisy samples are present, which may result in inaccurate assessments of a sample's effect. For instance, some methods employ difficulty-based criteria to select data; however, distinguishing between truly difficult samples and noisy ones based solely on image modalities presents a significant challenge. On the other hand, existing methods typically select samples with the highest or lowest scores, while the interaction between high-score and low-score samples within a group can significantly influence the overall performance, which is known as the "group effect" Koh et al. (2019); Yang et al. (2023b). Thus, a more beneficial approach is to leverage the power of multimodal information and evaluate the collective effect of the sample group.

In this paper, we propose a CLIP-powered data selection framework that employs multimodal information for more robust and generalizable data selection, where the category text serves as a strong complement to the image modality and promotes overall performance. The framework consists of three modules, namely dataset adaptation, sampling scoring, and selection optimization module. First, the dataset adaptation module integrates image and text adapters to facilitate the transfer of pretraining knowledge to the target data. Subsequently, the sample scoring module calculates the Semantic Alignment Score (SAS) and Sample Diversity Score (SDS) based on the adapted multimodal features, which measure the image-text alignment and the variability of local patterns. Using these two scores together can select semantically representative samples while maintaining their inherent diversity. Further, in order to address the group effect, we introduce a selection optimization module to identify the ideal subsets w.r.t. the expected selection ratio through a multi-objective optimization strategy. By leveraging the multi-modal information and carefully designed supervision signals, our framework enables the selection of high-quality samples in a flexible and efficient manner.

Comprehensive evaluation across various benchmark datasets demonstrates that our approach effectively improves the performance of selected datasets, especially on large-scale datasets such as ImageNet-1k Deng et al. (2009). Moreover, the selected datasets exhibit superior cross-architecture generalization across ResNet-18/50 He et al. (2016), ViT Dosovitskiy et al. (2020), VGG-16 Simonyan & Zisserman (2014), etc. Notably, since most existing methods are not robust to more complex and realistic scenarios, we further validate the strong robustness of our method in more challenging scenes. For instance, our proposed method can achieve an 8.13% improvement in accuracy on CIFAR-100 and a 4.41% improvement on Tiny-ImageNet compared to the leading baselines. Meanwhile, superior performance is achieved with very high efficiency compared to other baselines.

The contributions of this work can be summarized as follows: 1) We analyze the drawbacks of previous works that rely solely on image modalities in depth, and propose a new CLIP-powered data selection framework that leverages multimodal features for robust and generalizable data selection

for the first time. 2) Our framework comprises dataset adaptation and sample scoring modules to foster multi-modality knowledge transfer and comprehensive sample importance evaluation. This dual-modality design effectively removes noisy and corrupted samples from the dataset. 3) A selection optimization module is designed to identify the optimal subsets w.r.t. the expected selection ratios through multi-objective optimization, which effectively addresses the group effect while maintaining high efficiency. 4) Experimental results show that our method outperforms previous SOTA approaches in terms of performance, cross-architecture generalization, and robustness to noisy and corrupted images. Meanwhile, our approach achieves the best trade-off in performance and selection efficiency, establishing a strong baseline of data selection for future research.

## 2 RELATED WORK

Date-efficient deep learning generally incorporates dynamic data pruning Qin et al. (2024); Raju et al. (2021), static data selection Xia et al. (2023); Tan et al. (2024); Yang et al. (2023b), data distillation Lei & Tao (2023); Du et al. (2023); Zhang et al. (2023); Cazenavette et al. (2022), and data condensation Liu et al. (2023); Yang et al. (2023a). Following the static data selection, we propose a method capable of identifying representative and diverse samples across various selection ratios. Data selection, or static dataset pruning, aims to identify and select the most representative samples from training datasets before training begins. Training on the selected subsets can achieve comparable performance to that obtained with the full dataset while reducing training and storage costs. Current data selection methods can be broadly divided into three categories: importance criteria Paul et al. (2021); Tan et al. (2024), dataset distribution-based methods Xia et al. (2023); Zheng et al. (2023), and optimization-based methods Nohyun et al. (2023); Yang et al. (2023c).

**Selection with importance criteria** is the most popular type. These methods typically involve calculating importance scores for each sample and selecting samples based on these scores. For instance, EL2N and GraNd score Paul et al. (2021) measure the importance by calculating the expectation of the $\ell_2$-norm error vector and the expectation of the gradient norm, respectively. MoSo Tan et al. (2024) calculates the change of the optimal empirical risk when removing a specific sample from the training set. Forgetting Toneva et al. (2018) tracks the frequency with which a sample is misclassified after being correctly classified during training. Similarly, Memorization Feldman & Zhang (2020) assesses the impact of a sample's presence or absence on the model's ability to predict it correctly. While importance criteria-based methods are often computationally efficient, their performance may be affected by the group effect Yang et al. (2023b;c) and may not generalize well to complex, real-world scenarios Xia et al. (2023).

**Dataset distribution-based methods** select samples by analyzing the geometric distribution of datasets. For instance, Herding Welling (2009) determines the sample importance according to samples' distance to their corresponding class centers. The work Ramalingam et al. (2023) applies greedy k-center to select the coreset with good data coverage. D2 Maharana et al. (2023) calculates and updates the difficulty scores of each sample by incorporating the difficulty of its neighboring examples. Moderate-DS Xia et al. (2023) choose samples with closer distances to the median score, while CCS Zheng et al. (2023) balances the data distribution and the sample importance in selection.

**Optimization-based methods** select samples through various optimization techniques, such as temporal dual-depth scoring Zhang et al. (2024), gradient matching Mirzasoleiman et al. (2020b); Killamsetty et al. (2021a), scalable self-supervised pruning metric Sorscher et al. (2022), influence function Yang et al. (2023b); Pooladzandi et al. (2022), bi-level optimization Killamsetty et al. (2021b), facility location function Mirzasoleiman et al. (2020a); Yang et al. (2023d), and submodularity Iyer et al. (2021); Nohyun et al. (2023); Kothawade et al. (2022); Wei et al. (2015). In contrast to these methods that only rely on image information, we leverage multimodal messages for data selection, which incorporates the semantic alignment between image data and corresponding category information. This contributes to comprehensive assessments of sample effectiveness, particularly in complex scenarios where samples may be corrupted or wrongly labeled.

## 3 THE PROPOSED METHOD

Our proposed method is summarized in Figure 2. The approach involves the use of the pretrained vision-language foundation model CLIP to construct multimodal feature spaces. Nevertheless,

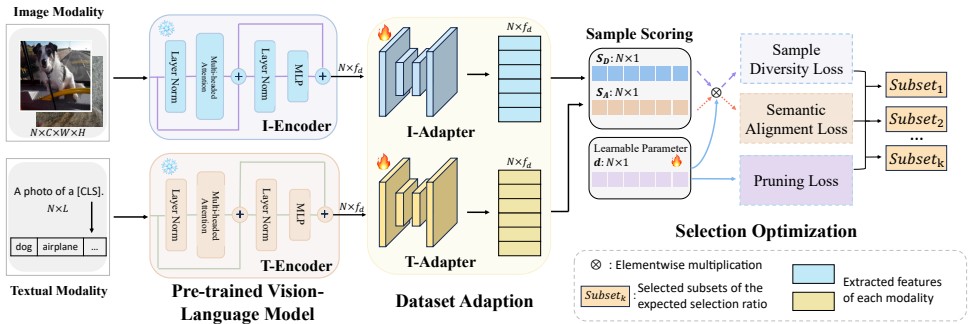

Figure 2: Our proposed method consists of dataset adaptation, sampling scoring, and selection optimization modules. The dataset adaptation module is used to learn dataset-specific knowledge. The sample scoring module computes two scores, $\boldsymbol{S}_A$ and $\boldsymbol{S}_D$, to assess sample importance, based on which the selection optimization identifies the optimal subsets w.r.t. the expected selection ratios.

there may exist domain shifts or discrepancies between the pretrained training dataset and the target dataset Liu et al. (2024a); Alijani et al. (2024). To facilitate dataset adaptation and enhance the learning of dataset-specific knowledge, we incorporate dimension-preserving adapters for both modalities. Following this, two scores are derived to comprehensively assess the sample importance, i.e., the Semantic Alignment Score (SAS), denoted as $\boldsymbol{S}_A$, and the Sample Diversity Score (SDS), denoted as $\boldsymbol{S}_D$. Furthermore, rather than solely based on sample scores for selection, we design a multi-objective optimization to identify the optimal subsets w.r.t. the expected selection ratios, which effectively mitigates the group effect. We provide the detailed methodology in the subsequent sections.

## 3.1 DATASET ADAPTATION

To alleviate the domain shifts and discrepancies between the pretrained and target datasets, we incorporate dimension-preserving adapters to perform adaptation on the target dataset. The image and text adapters are denoted as $A_I$ and $A_T$, respectively. Both adapters are fine-tuned for knowledge transfer while the pretrained CLIP weights are frozen. To maintain high efficiency, both adapters utilize simple MLP.

Specifically, the fine-tuning process employs the InfoNCE loss Parulekar et al. (2023); Oord et al. (2018), which maximizes the mutual information between the image and text representations. The text representation describes the category information using the prompt: "A photo of [CLS]", where the token [CLS] represents the corresponding category. The loss ensures that the adapters effectively align and capture the relevant features from both modalities. Furthermore, it enhances the model's ability to distinguish between positive and negative pairs, thereby improving the robustness and accuracy of the deep representations for the specific dataset.

## 3.2 SAMPLE SCORING

For classification datasets, the learning process for training samples is intrinsically linked to acquiring knowledge of the corresponding categories. A training sample that more accurately represents its category is typically more effective in training deep networks. In this way, the **Semantic Alignment Score (SAS)** is designed to assess the semantic similarity between training samples and their corresponding categories. Specifically, since image and text features reside within the same embedding space Radford et al. (2021), the SAS is derived by calculating the cosine similarity between the embedded image and corresponding textual deep descriptions. Accordingly, the SAS for the $i$-th sample is defined as:

$$\boldsymbol{S}_{Ai} = \cos(A_I(E_I(\boldsymbol{I}_i)), A_T(E_T(\boldsymbol{T}_i))), \tag{1}$$

where $\boldsymbol{I}_i$ is the $i$-th sample, $\boldsymbol{T}_i$ is the textual description of the corresponding category for $\boldsymbol{I}_i$, $E_I$ and $E_T$ are pretrained image and text encoders, respectively.

For selected datasets, the reduced data volume may limit the diversity of the selected data, which is important for training datasets Yang et al. (2024a). To address this, we introduce another diversity

Figure 3: Illustration of the effectiveness of SAS and SDS. The circle and cross represent the normal and noisy samples, respectively. Different colors correspond to the selection results. SDS (b) selects diverse samples but may include noises. SAS (c) could avoid the noisy samples but potentially miss broader category information. Using both scores (d) can select category-representative while maintaining high diversity.

perspective to comprehensively assess the effect of training data. The **Sample Diversity Score (SDS)** is defined as the average distance between each sample and its $k$ neighbor samples of the same class:

$$\boldsymbol{S}_{Di} = \frac{1}{k} \sum_{\boldsymbol{I}_j \in \text{KNN}(\boldsymbol{I}_i)} \|A_I(E_I(\boldsymbol{I}_i)) - A_I(E_I(\boldsymbol{I}_j))\|, \tag{2}$$

where we use the KNN algorithm to obtain the neighbor samples for each sample, the distance metric is based on the $\ell_2$ norm, and $k$ is usually set to 10% of the number of samples per class. In this way, the SDS can be understood as the local density of training samples in the feature space. If a sample has a larger number of neighbors with closer distances (i.e., lower SDS), its training efficacy may be more easily substituted by other neighbor samples. Therefore, selecting samples with higher SDS is generally more advantageous.

The effects of SAS and SDS are depicted in Figure 3. SDS contributes to the diversity of samples, but the selected data points may contain noise (Figure 3(b)). On the other hand, SAS can select samples that are semantically appropriate, as these points are basically around the sample center (Figure 3(c)). However, these samples may be too concentrated and thus lack diversity. When using SDS and SAS together (Figure 3(d)), we can cover the entire category space with fewer data and select samples that are both semantically representative and diverse, thereby boosting the effectiveness of data selection.

### 3.3 SELECTION OPTIMIZATION

Instead of relying on combinatorial optimization functions for sample selection Killamsetty et al. (2021b), our method determines the selection through SGD multi-objective optimization, which improves computational efficiency and accelerates convergence. Specifically, we introduce a sample-wise parameter $\boldsymbol{d}$ to denote the selection decision, where elements of 1 indicate the selection while 0 indicates otherwise. Although binary parameters are difficult to optimize in neural networks due to the absence of gradient, we employ the $\text{sigmoid}(\cdot)$ function to push the continuous values of $\boldsymbol{d}$ towards approximate binarization. After optimization, $\boldsymbol{d}$ is strictly binarized to explicitly indicate the final sample selection. Initially, $\boldsymbol{d}$ is initialized with all 1s.

To guide the optimization process, we introduce three loss items. The first item, $\mathcal{L}_{sa}$, is designed to prioritize samples with high SAS since these samples are more representative of their corresponding categories, which is defined as follows:

$$\mathcal{L}_{sa} = -\frac{1}{N} \sum_i^N \text{sigmoid}(\boldsymbol{d}) * \boldsymbol{S}_{Ai}, \tag{3}$$

where $N$ is the total number of samples. $\mathcal{L}_{sa}$ punishes samples with low SAS and encourages the selection of samples with better semantic alignment.

In addition, we introduce another loss term, $\mathcal{L}_{sd}$, to encourage the selection of more diverse samples characterized by higher SDS, which is defined as:

$$\mathcal{L}_{sd} = -\frac{1}{N} \sum \text{sigmoid}(\boldsymbol{d}) * \boldsymbol{S}_{Di}. \tag{4}$$

To mitigate the group effect, we optimize the selected datasets w.r.t. specific selection ratios, aiming to identify the optimal subsets. We introduce a selection loss term, $\mathcal{L}_s$, to ensure the selection

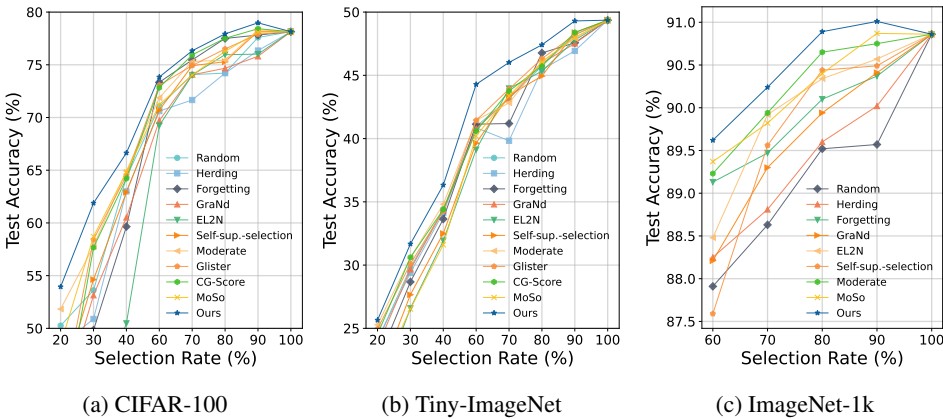

|                |                |                  |
|:--------------:|:--------------:|:----------------:|
| (a) CIFAR-100  | (b) Tiny-ImageNet | (c) ImageNet-1k |

Figure 4: Illustrations of comparing our method with various data selection baselines on CIFAR-100 (a), Tiny-ImageNet (b), and ImageNet-1k (c).

process adheres to the target ratio. However, deriving exact selection rates from continuous real-valued parameter optimization is difficult. While strictly binarized values facilitate explicit sample selection, they are challenging to optimize through gradient backpropagation. To address this, we utlize the straight-through estimator (STE) Bengio et al. (2013) to estimate the actual selection rate and derive gradients. STE allows gradients to pass through the discrete decisions during backprop-agation, effectively combining the benefits of both continuous and binary parameters for efficient optimization and accurate sample selection. In this way, $\mathcal{L}_s$ is defined as:

$$\mathcal{L}_s = \sqrt{\left[\frac{1}{N}\sum_i \text{STE}\left[\mathbb{I}\left(\text{sigmoid}(\boldsymbol{d}_i)_{\text{i}} > 0.5\right)\right] - s_r\right]^2}, \tag{5}$$

where $\mathbb{I}$ is an indicator function, and $s_r$ denotes the expected selection ratio. The loss term guides the parameter $\boldsymbol{d}$ toward near-binary values, ensuring the count of ones aligns with the expected sample size. Since the selection is guided by adaptive optimization, the final selection ratio may deviate slightly from the target. To minimize this deviation, we constrain $\mathcal{L}_s$ with a threshold $\theta$, which in our work is set to $5 \times 10^{-4}$, ensuring the actual selection ratio differs by less than $\pm 0.05\%$ from the expected value. We also provide a theoretical analysis of $\theta$ on the actual selection ratio gap in Appendix A. Finally, the overall loss function is formulated as:

$$\mathcal{L} = \mathcal{L}_{sa} + \alpha\mathcal{L}_{sd} + \beta\mathcal{L}_s, \tag{6}$$

where $\alpha$ and $\beta$ are coefficients that adjust for numerical differences among the loss terms and can be set conveniently. The complete workflow is outlined in Algorithm 1 in Appendix B.

**Complexity Analysis** The proposed method comprises three main components. 1) The dataset adaptation involves fine-tuning the image and text adapters. Since the adapters consist of simple lin-ear layers, the number of parameters is small, and both the forward passes and backward passes are computationally efficient. 2) In the sample scoring process, the complexity of calculating the SAS and SDS is $O(N)$ and $O(K * f_d)$, respectively, where $K$ is the number of categories and $f_d$ is the feature dimension, typically 512. The complexity of the KNN algorithm is $O(N_k * f_d)$, where $N_k$ is the number of samples per class. Given that $K$ and $f_d$ are constants and $N_k$ is usually much smaller than $N$, the overall complexity of this process is approximately $O(N)$. 3) The selection optimiza-tion of $\boldsymbol{d}$ does not involve deep models and is a numerical optimization process. The complexity is proportional to the number of parameters, i.e., $O(|w|) = O(N)$.

## 4 EXPERIMENT

### 4.1 EXPERIMENTAL SETUP

**Baselines.** We compare our proposed method with ten most representative SOTA baselines, i.e., (1) Random, (2) MoSo Tan et al. (2024), (3) Glister Killamsetty et al. (2021b), (4) Herding Welling

Table 1: Test accuracy (%) on Tiny-ImageNet. VGG-16 and DenseNet-121 are exploited.

| Method / Selection Ratio | VGG-16 | | | | Densenet-121 | | | |
|---|---|---|---|---|---|---|---|---|
| | 70% | 80% | 90% | 100% | 70% | 80% | 90% | 100% |
| Random | $47.39_{\pm2.72}$ | $49.38_{\pm0.23}$ | $51.15_{\pm0.64}$ | $57.23_{\pm1.08}$ | $59.55_{\pm0.20}$ | $60.78_{\pm0.18}$ | $61.03_{\pm0.22}$ | $62.22_{\pm0.23}$ |
| EL2N | $48.30_{\pm2.95}$ | $48.75_{\pm1.65}$ | $49.01_{\pm1.31}$ | $57.23_{\pm1.08}$ | $59.61_{\pm0.00}$ | $60.38_{\pm0.04}$ | $61.16_{\pm0.47}$ | $62.22_{\pm0.23}$ |
| GraNd | $50.79_{\pm1.26}$ | $46.84_{\pm1.38}$ | $54.73_{\pm0.49}$ | $57.23_{\pm1.08}$ | $59.62_{\pm0.02}$ | $60.84_{\pm0.09}$ | $61.10_{\pm0.05}$ | $62.22_{\pm0.23}$ |
| MoSo | $50.47_{\pm1.01}$ | $50.12_{\pm0.83}$ | $50.07_{\pm0.43}$ | $57.23_{\pm1.08}$ | $59.27_{\pm0.33}$ | $59.86_{\pm0.07}$ | $60.00_{\pm0.37}$ | $62.22_{\pm0.23}$ |
| Herding | $48.59_{\pm0.07}$ | $45.77_{\pm0.12}$ | $50.77_{\pm1.24}$ | $57.23_{\pm1.08}$ | $59.00_{\pm0.28}$ | $60.03_{\pm0.35}$ | $61.15_{\pm0.12}$ | $62.22_{\pm0.23}$ |
| Glister | $48.74_{\pm2.29}$ | $50.05_{\pm0.02}$ | $49.42_{\pm1.81}$ | $57.23_{\pm1.08}$ | $59.98_{\pm0.01}$ | $60.62_{\pm0.34}$ | $61.28_{\pm0.18}$ | $62.22_{\pm0.23}$ |
| CG-Score | $48.73_{\pm2.70}$ | $48.49_{\pm1.88}$ | $49.62_{\pm1.08}$ | $57.23_{\pm1.08}$ | $59.74_{\pm0.15}$ | $60.55_{\pm0.20}$ | $61.14_{\pm0.11}$ | $62.22_{\pm0.23}$ |
| Self-sup. prototypes | $48.38_{\pm1.38}$ | $49.98_{\pm1.49}$ | $54.71_{\pm0.84}$ | $57.23_{\pm1.08}$ | $59.56_{\pm0.03}$ | $60.22_{\pm0.12}$ | $60.91_{\pm0.29}$ | $62.22_{\pm0.23}$ |
| Forgetting | $47.50_{\pm2.43}$ | $48.59_{\pm1.77}$ | $49.82_{\pm0.62}$ | $57.23_{\pm1.08}$ | $58.54_{\pm0.15}$ | $60.39_{\pm0.46}$ | $61.12_{\pm0.10}$ | $62.22_{\pm0.23}$ |
| Moderate-DS | $50.78_{\pm0.93}$ | $49.31_{\pm0.41}$ | $49.25_{\pm0.77}$ | $57.23_{\pm1.08}$ | $59.41_{\pm0.18}$ | $60.42_{\pm0.14}$ | $61.44_{\pm0.11}$ | $62.22_{\pm0.23}$ |
| Ours | $\mathbf{53.40_{\pm3.20}}$ | $\mathbf{52.25_{\pm0.58}}$ | $\mathbf{56.34_{\pm2.93}}$ | $57.23_{\pm1.08}$ | $\mathbf{60.12_{\pm0.06}}$ | $\mathbf{60.93_{\pm0.03}}$ | $\mathbf{61.59_{\pm0.03}}$ | $62.22_{\pm0.23}$ |

Table 2: Experimental results (accuracy, %, mean ± std) on CIFAR-100 and Tiny-ImageNet with noisy labels. 20% of labels are disturbed. We also report the numerical analysis of the proportion (%) of noisy data in the selected CIFAR-100 datasets.

| Method / Selection Ratio | CIFAR-100 (label noise) | | Tiny-ImageNet (label noise) | | Noise ratios | |
|---|---|---|---|---|---|---|
| | 20% | 30% | 20% | 30% | 20% | 30% |
| Random | $34.47_{\pm0.64}$ | $43.26_{\pm1.21}$ | $17.78_{\pm0.44}$ | $23.88_{\pm0.42}$ | 20.80 | 19.83 |
| MoSo | $31.01_{\pm0.67}$ | $43.73_{\pm0.14}$ | $21.55_{\pm0.37}$ | $27.80_{\pm0.16}$ | 7.78 | 8.82 |
| Moderate-DS | $40.25_{\pm0.12}$ | $48.53_{\pm1.60}$ | $19.64_{\pm0.40}$ | $24.96_{\pm0.30}$ | 0.30 | 0.31 |
| Glister | $28.51_{\pm1.46}$ | $43.16_{\pm1.31}$ | $21.61_{\pm0.19}$ | $25.45_{\pm0.23}$ | 21.21 | 21.95 |
| Herding | $42.29_{\pm1.75}$ | $50.52_{\pm3.38}$ | $18.98_{\pm0.44}$ | $24.23_{\pm0.29}$ | 35.00 | 30.56 |
| Forgetting | $36.53_{\pm1.11}$ | $45.78_{\pm1.04}$ | $13.20_{\pm0.38}$ | $21.79_{\pm0.43}$ | 23.00 | 21.76 |
| GraNd | $31.72_{\pm0.67}$ | $42.80_{\pm0.30}$ | $18.28_{\pm0.32}$ | $23.72_{\pm0.18}$ | 5.00 | 5.14 |
| EL2N | $29.82_{\pm1.19}$ | $33.62_{\pm2.35}$ | $13.93_{\pm0.69}$ | $18.57_{\pm0.31}$ | 22.00 | 21.80 |
| Self-sup. prototypes | $31.08_{\pm0.78}$ | $41.87_{\pm0.63}$ | $15.10_{\pm0.73}$ | $21.01_{\pm0.36}$ | 21.70 | 20.21 |
| CG-Score | $6.82_{\pm1.60}$ | $20.07_{\pm0.45}$ | $8.35_{\pm0.65}$ | $15.31_{\pm0.90}$ | 45.09 | 39.69 |
| Ours | $\mathbf{46.05_{\pm0.21}}$ | $\mathbf{58.34_{\pm0.36}}$ | $\mathbf{26.09_{\pm0.12}}$ | $\mathbf{33.13_{\pm0.25}}$ | $\mathbf{0.25}$ | $\mathbf{0.32}$ |

(2009), (5) Forgetting Toneva et al. (2018), (6) GraNd and (7) EL2N Paul et al. (2021), (8) Self-sup.-selection (SSP) Sorscher et al. (2022), (9) CG-Score Nohyun et al. (2023), and (10) Moderate-DS Xia et al. (2023).

**Parameter settings.** The parameters in our proposed method can be easily set. The coefficient $\alpha$ is proportional to the expected selection rate $s_r$, balancing the importance of dataset diversity, i.e., $\alpha$ can be set equivalent to $s_r$. The coefficient $\beta$ is set to 2 across datasets to adjust the numerical differences among loss items. For more details, please refer to Appendix C.

## 4.2 COMPARISON WITH THE STATE-OF-THE-ARTS

**Performance Comparisons**    Consistent with prior works Xia et al. (2023); Sorscher et al. (2022), we report top-1 accuracy on CIFAR-100 and Tiny-ImageNet, and top-5 accuracy on ImageNet-1k. Note that the methods Glister and CG-Score are not compared on ImageNet-1k due to the heavy computation costs. Specifically, Glister obtains iterative solving of the bi-level optimization problem Killamsetty et al. (2021b), while CG-Score involves the calculation of large Gram matrix inversions, making them impractical for large-scale datasets.

As illustrated in Figure 4, our method consistently achieves the best accuracy across all datasets. Particularly, on the more challenging Tiny-ImageNet and ImageNet-1k datasets, our approach outperforms other methods by a notable margin. While existing approaches yield relatively marginal accuracy improvements on the small-scale CIFAR-100 dataset, the gains brought by our method are more substantial. Additionally, with relatively high selection ratios on these datasets, such as 90%, our selected datasets exhibit nearly lossless or even higher performance compared with the original full datasets and other baselines. The results indicate that our method can not only reduce training costs but may also serve as a way for data cleaning.

**Generalization Comparisons on Different Architectures**    In this section, we evaluate the generalization effectiveness of our selected datasets on deep architectures different from those used in the selection process. Specifically, we employ the VGG-16 and DenseNet-121 models to train on the selected datasets from Tiny-ImageNet. As shown in Table 1, the results indicate that our method surpasses all baseline methods on both architectures, demonstrating its desirable architectural gen-

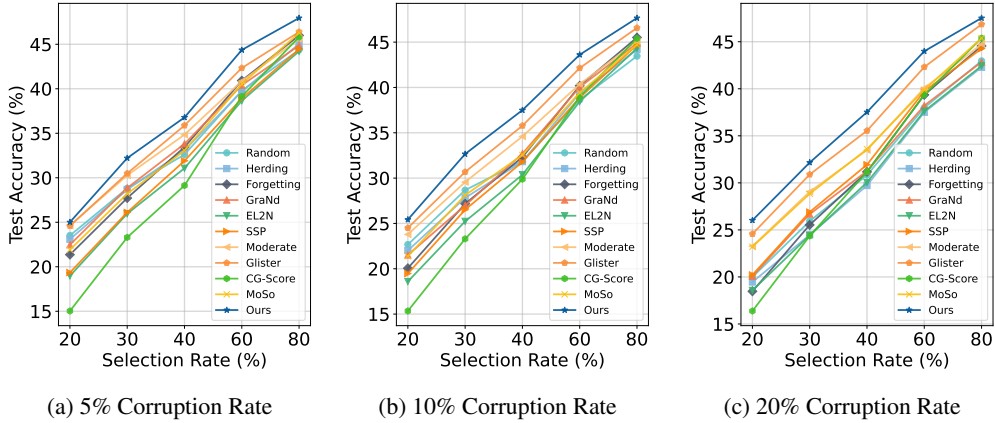

|  |  |  |
|---|---|---|
| (a) 5% Corruption Rate | (b) 10% Corruption Rate | (c) 20% Corruption Rate |

Figure 5: Comparison of robustness to corrupted images.

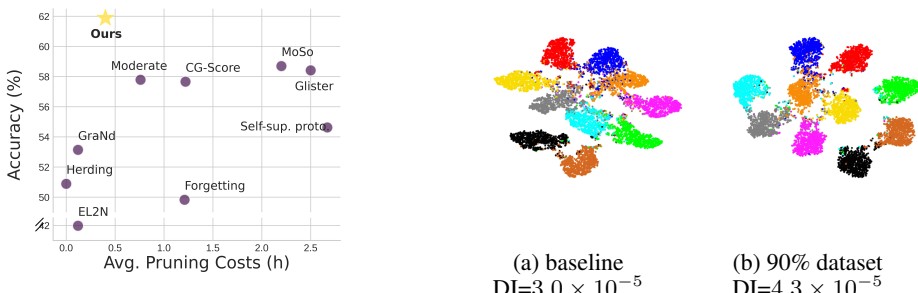

|  |  |
|---|---|
|  | (a) baseline $DI=3.0 \times 10^{-5}$ |
|  | (b) 90% dataset $DI=4.3 \times 10^{-5}$ |

Figure 6: Effectiveness *vs.* efficiency on C-100. Results are reported with R-50 under 30% selection ratio on a 4-2080TI GPU.

Figure 7: Visualization of the test set distribution. DI: Dunn Index.

eralization ability. This suggests that the selected datasets are broadly applicable, irrespective of the specific network architecture.

**Training Efficiency Comparisons**    To evaluate the selection efficiency of various methods, we present an analysis of the balance between effectiveness and efficiency. As shown in Figure 6, our method presents the best performance with desirable efficiency. Herding, EL2N, and GraNd obtain the lowest selection costs because they rely on predefined metrics or select samples in very early training. Our method is slightly slower than theirs but exhibits higher accuracy. Compared with the optimization-based approaches, like MoSo and Glister, our method enjoys both lower costs and better performance. The results verify the effectiveness of our method in balancing selection efficiency and accuracy.

## 4.3    COMPARISON OF ROBUSTNESS

**Robustness on Noisy Labels**    Real-world datasets often involve label noise, where some sample labels are incorrectly flipped, resulting in mislabeled data. Unfortunately, creating clean and diverse datasets is time-consuming and expensive. Therefore, it is necessary to evaluate the performance of selection methods under such complex scenarios. In this study, we introduce symmetric noises Li et al. (2022) to generate mislabeled data on both C-100 and T-ImageNet, with a 20% noise rate.

As can be seen in Table 2, our approach exhibits superior robustness to noisy labels, significantly outperforming other baselines by a large margin. Specifically, our approach yields improvements of over 10.12% on CIFAR-100 and 4.41% on Tiny-ImageNet compared to previous leading methods. Additionally, Table 2 shows the distribution of noisy data within the selected datasets, where our method selects only 0.24% noisy samples, considerably fewer than other baselines. This reduction in noise underscores the potential of our method to improve overall data quality.

Table 3: Performance and saved costs (%) on ImageNet-1k across Swin-T, ViT-B, and ViT-L on a 4-A100-GPU serve.

| Model | 80% | 90% | Full Data |
|---|---|---|---|
| ViT-B | 81.13 | **81.46** | 81.46 |
| ViT-L | 84.37 | **84.74** | 84.59 |
| Swin-T | 78.05 | **78.63** | 78.31 |
| Saved (%) | 20.62% | 10.31% | - |

Table 4: Generalization evaluation on more challenging ImageNet-1k benchmark datasets.

| Dataset | Model | 80% | 90% | Full Data |
|---|---|---|---|---|
| ImageNet-Hard | R-18 | 10.89 | **11.33** | 10.85 |
|  | R-50 | 14.75 | **14.98** | 14.75 |
| ImageNet-A | R-18 | 1.65 | **2.04** | 1.12 |
|  | R-50 | 3.17 | **3.31** | 3.09 |
| ImageNet-R | R-18 | 32.99 | **33.70** | 33.03 |
|  | R-50 | 36.60 | **37.11** | 36.16 |

We argue that the robustness of our approach can be attributed to $S_A$ in Eq. 1, which assesses the semantic alignment between image content and its labels. In the presence of label noise, this alignment is disrupted, resulting in a lower SAS, which in turn reduces the likelihood of such samples being selected during optimization. In contrast, most baseline methods rely solely on image features for selection, which may result in performance degradation when faced with noisy labels. In some cases, the performance of these methods is even worse than that of random selection. While Moderate also selects a relatively low proportion of noisy data, its performance is worse than ours. This discrepancy highlights the effectiveness of our method in making more strategic selections in noisy environments, thereby not only minimizing noises but also optimizing the selected datasets.

**Robustness on Corrupted Images** We further evaluate the performance of our proposed method on real-world noise corruptions that are frequently encountered Singh et al. (2024); Wei et al. (2024). To simulate such corruptions, we employ the following five types of realistic noises Xia et al. (2023), including Gaussian noise, random occlusion, resolution, fog, and motion blur. The corruption rate is set to 5%, 10%, and 20%, respectively.

As shown in Figure 5, compared with prior baselines, our approach consistently presents greater robustness to corrupted images across varying corruption rates, demonstrating strong generalization in these challenging scenarios. Notably, even at a high corruption rate of 20%, our method maintains desirable generalization performance. This robustness is primarily attributed to the integration of text modality into the selection process, alongside the image modality. The SAS defined in Eq. 1 measures the alignment between the image features and their corresponding category features. When images are corrupted, this alignment is disrupted, thereby reducing the SAS and correspondingly decreasing the likelihood of selecting those images. In contrast, methods such as Forgetting tend to prioritize difficult training samples, potentially making corrupted images more likely to be selected, as these images are typically more difficult to correctly classify. As a result, these methods are less robust to corrupted images, leading to a deterioration in generalization performance.

## 4.4 DATASET SELECTION IMPROVES GENERALIZATION

**Visualization Analysis** To demonstrate the generalization of the selected datasets, we train two models using the original dataset and the selected dataset (90% selection ratio), respectively, and obtain their embedding results on the CIFAR-10 test set. To visualize the dataset distribution, we apply t-SNE to the embeddings generated by both models. The visualization in Figure 7 shows that the model trained on the selected dataset produces better embedding results: a better inter-cluster separation and intra-cluster compactness. For a quantitative analysis, we use the Dunn Index (DI) Ncir et al. (2021) to evaluate the clustering results (the higher, the better). After removing 10% of the data, the DI increases by 43%, presenting better clustering results.

**Generalization to More Advanced Architectures** We further employ the selected datasets to train more advanced ViT-based architectures, including Swin Transformer, ViT-Base, and ViT-Large. From Table 3, and corroborated by the results from previous sections, our selected datasets consistently achieve lossless performance across both CNN-based and Transformer-based architectures with reduced training costs. This demonstrates that our approach obtains highly generalizable datasets applicable to a wide range of network architectures.

**Generalization to More Challenging Benchmark Datasets** To further evaluate the generalization and robustness of models trained on our selected datasets, we conduct experiments using ResNet-18 and ResNet-50 models, training on both the full datasets and our selected datasets. These models are then tested on more challenging benchmarks, including ImageNet-Hard Taesiri

Table 5: Evaluation of our components on CIFAR-100 (C-100) and Tiny-ImageNet (T-IN).

| | w/o adapter | w/o $\mathcal{L}_{sa}$ | w/o $\mathcal{L}_{sd}$ | w/o $\mathcal{L}_s$ | w/o adp&$\mathcal{L}_{sa}$ | w/o adp&$\mathcal{L}_{sd}$ | w/o adp&$\mathcal{L}_s$ | w/o adp&$\mathcal{L}_{sd}$&$\mathcal{L}_{sa}$ | Ours |
|---|---|---|---|---|---|---|---|---|---|
| C-100 | $78.20_{\pm 0.18}$ | $78.42_{\pm 0.46}$ | $78.85_{\pm 0.05}$ | $78.48_{\pm 0.32}$ | $78.11_{\pm 0.16}$ | $78.21_{\pm 0.07}$ | $77.10_{\pm 0.29}$ | $77.47_{\pm 0.31}$ | $\mathbf{78.98_{\pm 0.09}}$ |
| T-IN | $46.68_{\pm 0.12}$ | $46.79_{\pm 0.39}$ | $49.14_{\pm 0.09}$ | $46.01_{\pm 0.38}$ | $47.23_{\pm 0.06}$ | $46.70_{\pm 0.33}$ | $45.79_{\pm 0.11}$ | $45.69_{\pm 0.10}$ | $\mathbf{49.30_{\pm 0.12}}$ |

et al. (2024), ImageNet-R Hendrycks et al. (2021a), and ImageNet-A Hendrycks et al. (2021b). The results, shown in Table 4, demonstrate that models trained on our selected data consistently exhibit superior generalization and robustness on these harder ImageNet benchmarks compared to those trained on the original datasets. Notably, this improved performance is achieved with reduced training costs, further highlighting the efficacy of our approach.

## 4.5 ABLATION STUDY

**Effect of Dataset Adaptation**     To assess the effect of dataset adaptation, instead of using fine-tuned image and text adapters, we directly utilize the per-trained CLIP model to derive the scores $S_A$ and $S_D$. The experimental results, presented in Table 5 with a 90% selection ratio, show a significant decline in accuracy, with drops exceeding 2% on Tiny-ImageNet. Thus, dataset adaptation is essential for effectively transferring the model's generalization ability to target datasets. This is particularly crucial for datasets that differ substantially from the pre-training datasets, such as CIFAR, where variations in image sizes and domain are distinct.

**Effect of Text Modality**     Previous works Xia et al. (2023) use the average image features as the prototype and calculate the Euclidean distance between the embedded image and the corresponding prototype, which is used to filter noisy labels Wu et al. (2021). To evaluate the effect of our text modality, we also leverage this distance to assess the noisy samples and replace the text feature in Eq. 1. With a 20% noisy ratio, the accuracy drops from 46.05% to 16.39% with a 20% selection ratio and from 58.34% to 38.35% with a 30% selection ratio, validating the effectiveness of introducing textual modality. Due to the limited space, full results can be seen in Appendix L.

**Effect of Loss Terms**     In Table 5, we evaluate the effect of each loss term and their combination in Eq. 6. The overall loss function achieves the highest accuracy. When $\mathcal{L}_{sa}$ is omitted, the selection process tends to prefer more diverse samples, but some class-representative samples may not be selected, which deteriorates the model performance severely. Without $\mathcal{L}_{sd}$, the selection emphasizes the most category-representative samples. Although the resulting performance drop is slightly smaller, the diversity of the selected datasets is compromised. Therefore, the incorporation of $\mathcal{L}_{sd}$ ensures a balanced representation of the selected dataset. Without $\mathcal{L}_s$, since we can not obtain the binarized selection decisions w.r.t. the expected selection ratios, we directly sort the scores in $d$ and select the samples with higher scores. However, this degrades our method into a totally score-based selection and fails to address the group effect, leading to a noticeable drop in performance.

## 5 DISCUSSION AND CONCLUSION

**Limitation and future work.**     In this section, we discuss some potential limitations and future work for our method. 1). While the pretrained CLIP model contributes to selecting the most representative samples, the potential biases in the CLIP model may propagate to the selected dataset Alabdulmohsin et al. (2024). Future work may explore bias mitigation strategies by fine-tuning CLIP with bias-aware loss functions. 2). Our work primarily focuses on vision-based datasets for data selection, using textual modalities to guide sample selection, where both modalities are balanced. However, datasets with high modality imbalance may bring challenges since it is difficult to assess the modality alignment. Thus, future work could consider leveraging dynamic modality weight allocation mechanisms and augmentation strategies for modality-imbalanced dataset selection.

This paper proposes a novel CLIP-powered data selection framework that leverages multimodal information for more robust and generalizable sample selection. To achieve this, our proposed framework incorporates three modules: dataset adaptation, sample scoring, and selection optimization modules. These modules assess the data effectiveness in model training and optimize the selection results w.r.t. the expected results. As a result, our framework is capable of selecting the most representative samples with high diversity. Extensive experiments demonstrate the effectiveness and efficiency of our approach, especially in terms of generalization performance on large-scale datasets and robustness in more challenging scenarios, such as noisy and corrupted images.

## 6 ACKNOWLEDGEMENT

This work is supported by the STI 2030-Major Projects of China under Grant 2021ZD0201300, the Fundamental Research Funds for the Central Universities under Grant 2024300394, the National Natural Science Foundation of China under Grant 62276127. This work is supported by the Shanghai Municipal Science and Technology Major Project. This work is supported by Shanghai Artificial Intelligence Laboratory.

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

## A    ANALYSIS ON THE THRESHOLD $\theta$

We present the relationship between the value of $\theta$ and the actual selection ratio gap. Suppose that we have $\mathcal{L}_s = \sqrt{\left[\frac{1}{N}\sum_i \text{STE}\left[\mathbb{I}\left(\text{sigmoid}(\boldsymbol{d}_i)_i > 0.5\right)\right] - s_r\right]^2} \leq \theta$. Then, we have,

$$\frac{\|\boldsymbol{d}\|_1}{N} \leq s_r \pm \theta. \tag{7}$$

In this way, the actual selection ratio can be constrained into the $\theta$ gap. We set $\theta$ as $5 \times 10^{-4}$. For instance, on CIFAR datasets with 50000 samples, if the expected selected number is 40000, the actual selected number can be between 39975 and 40025.

## B    THE SPECIFIC ALGORITHM WORKFLOW

To better understand the workflow of our proposed method, we present the detailed algorithm in Algorithm 1 below.

---
**Algorithm 1** The general workflow.

---
**Input:** dataset $\mathcal{D}$, total number of training samples $N$, total number of epochs $T$, selection ratio $s_r$, a threshold $\theta$, the pretrained image and text encoders are $E_I$ and $E_T$, the fine-tuned image and text adapters are $A_I$ and $A_T$, respectively.

1: $\boldsymbol{d} \leftarrow \boldsymbol{1}$
2: $\boldsymbol{s} \leftarrow \boldsymbol{0}$
3: **for** i=0:N-1 **do**
4:     Calculate the SAS $\boldsymbol{S}_A$ according to Eq. 1
5:     Calculate the $K$ neighbors $\boldsymbol{x}'$ of $\boldsymbol{x}_i$ using the KNN algorithm
6:     Calculate the SDS $\boldsymbol{S}_D$ according to Eq. 2
7: **end for**
8: **for** t=0:T-1 **do**
9:     Calculate the loss $\mathcal{L}_{sa}$ according to Eq. 3
10:     Calculate the loss $\mathcal{L}_{sd}$ according to Eq. 4
11:     Calculate the pruning loss $\mathcal{L}_s$ according to Eq. 5
12:     Calculate the overall loss $\mathcal{L}$ according to Eq. 6
13:     Update $\boldsymbol{d}$ based on the loss $\mathcal{L}$ and SGD with momentum
14:     **if** $\mathcal{L}_s \leq \theta$ **then**
15:         break
16:     **end if**
17: **end for**
18: Return $\boldsymbol{d}$

---

## C    IMPLEMENTATION DETAILS

**Datasets and Network Architectures.**    Consistent with previous works Tan et al. (2024); Xia et al. (2023); Zheng et al. (2023), we evaluate the effectiveness of our proposed method on various popularly used benchmark datasets, including CIFAR-10/100 Krizhevsky et al. (2009), Tiny-ImageNet Chrabaszcz et al. (2017), and ImageNet-1k Deng et al. (2009). To evaluate the generalization performance of our selected datasets, we study the effectiveness of our proposed method on a wide range of network architectures, including ResNet-18/50 He et al. (2016), Vision Transformer (ViT) Dosovitskiy et al. (2020), Swin-Transformer Liu et al. (2021), VGG-16 Simonyan & Zisserman (2014), and DenseNet-121 Huang et al. (2017).

**Training on the Selected Datasets**    Closely following previous works Xia et al. (2023); Yang et al. (2023c); Sorscher et al. (2022), for experiments on CIFAR-10/100, we adopt a batch size of 128, an SGD optimizer with a momentum of 0.9, weight decay of $5e-4$, an initial learning rate of 0.1, and a total training epoch of 200. The learning rate is divided by 5 after the 60th, the 120th, and the 160th epoch. For experiments on Tiny-ImageNet, we adopt a batch size of 256, an SGD optimizer

with a momentum of 0.9, a weight decay of 1r-4, an initial learning rate of 0.1, and a total epoch of 90. The learning rate is divided by 10 after the 30th and the 60th epoch. For experiments on ImageNet-1k, following Xia et al. (2023); Sorscher et al. (2022); Yang et al. (2024b), the VISSL library Goyal et al. (2021) is exploited. We adopt a base learning rate of 0.01, a batch size of 256, an SGD optimizer with a momentum of 0.9, a weight decay of 1e-3, and a total epoch of 105. All experiments are conducted by three individual runs with different random seeds, while on ImageNet-1k, due to the huge training costs, the experiment in each case is performed once. Unless specified, the network architecture used is the ResNet-50 model. All hyperparameters and experimental settings for training on different selected datasets are kept the same.

**Fine-tuning the Adapters** We adopt the initial learning rate of $1 \times 10^{-4}$, the Adam optimizer with a step size of 30 epochs, a decay factor of 0.1, and a total epoch of 30. The batch size is set to 256 on CIFAR-10/100, 64 on Tiny-ImageNet, and 512 on ImageNet-1k.

**Selection Optimization** We adopt an SGD optimizer with a momentum of 0.9, an initial learning rate of $1 \times 10^{-3}$, and a total training iteration of $1 \times 10^{5}$. Since this optimization merely involves numerical optimization on parameter $d$ using vanilla SGD without introducing any deep networks, the optimization can be very efficiently completed.

# D    EXPERIMENTAL RESULTS ON CIFAR-10

Table 6: Test accuracy (%) on CIFAR-10 with ResNet-50.

| Method / Selection ratio | 20% | 30% | 40% | 60% | 70% | 80% | 90% | 100% |
|---|---|---|---|---|---|---|---|---|
| Random | $84.12_{\pm1.53}$ | $90.34_{\pm0.39}$ | $92.71_{\pm0.38}$ | $94.43_{\pm0.37}$ | $95.02_{\pm0.29}$ | $95.55_{\pm0.14}$ | $95.89_{\pm0.11}$ | $96.12_{\pm0.12}$ |
| EL2N | $70.32_{\pm0.74}$ | $87.48_{\pm0.80}$ | $89.23_{\pm0.61}$ | $94.43_{\pm0.27}$ | $95.17_{\pm0.27}$ | $95.55_{\pm0.18}$ | $96.01_{\pm0.20}$ | $96.12_{\pm0.12}$ |
| MoSo | $83.33_{\pm0.47}$ | $89.17_{\pm0.14}$ | $92.47_{\pm0.14}$ | $94.69_{\pm0.20}$ | $95.50_{\pm0.00}$ | $95.93_{\pm0.01}$ | $96.26_{\pm0.02}$ | $96.12_{\pm0.12}$ |
| GraNd | $79.23_{\pm0.84}$ | $87.88_{\pm0.90}$ | $92.17_{\pm0.73}$ | $94.14_{\pm0.47}$ | $95.19_{\pm0.12}$ | $95.35_{\pm0.38}$ | $95.96_{\pm0.05}$ | $96.12_{\pm0.12}$ |
| Glister | $79.23_{\pm0.55}$ | $87.88_{\pm0.49}$ | $92.17_{\pm0.34}$ | $95.03_{\pm0.13}$ | $95.61_{\pm0.05}$ | $95.98_{\pm0.17}$ | $96.34_{\pm0.02}$ | $96.12_{\pm0.12}$ |
| Herding | $78.42_{\pm0.78}$ | $87.77_{\pm0.66}$ | $89.40_{\pm0.54}$ | $89.12_{\pm0.35}$ | $92.11_{\pm0.13}$ | $93.92_{\pm0.36}$ | $95.50_{\pm0.13}$ | $96.12_{\pm0.12}$ |
| CG-Score | $80.50_{\pm1.23}$ | $89.35_{\pm0.87}$ | $92.73_{\pm0.37}$ | $95.19_{\pm0.23}$ | $95.87_{\pm0.17}$ | $95.99_{\pm0.16}$ | $96.16_{\pm0.15}$ | $96.12_{\pm0.12}$ |
| Forgetting | $67.58_{\pm1.05}$ | $88.12_{\pm1.40}$ | $93.61_{\pm0.87}$ | $95.17_{\pm0.25}$ | $95.85_{\pm0.20}$ | $95.46_{\pm0.27}$ | $95.85_{\pm0.37}$ | $96.12_{\pm0.12}$ |
| Moderate-DS | $81.75_{\pm0.38}$ | $90.94_{\pm0.27}$ | $92.79_{\pm0.31}$ | $94.69_{\pm0.24}$ | $95.26_{\pm0.30}$ | $95.73_{\pm0.19}$ | $96.17_{\pm0.15}$ | $96.12_{\pm0.12}$ |
| Self-sup. prototypes | $84.60_{\pm1.01}$ | $90.07_{\pm1.14}$ | $92.64_{\pm0.93}$ | $94.42_{\pm0.72}$ | $94.98_{\pm0.61}$ | $95.87_{\pm0.53}$ | $95.95_{\pm0.44}$ | $96.12_{\pm0.12}$ |
| Ours | $\mathbf{85.70}_{\pm0.06}$ | $\mathbf{91.10}_{\pm0.26}$ | $\mathbf{93.89}_{\pm0.13}$ | $\mathbf{94.85}_{\pm0.06}$ | $\mathbf{95.43}_{\pm0.21}$ | $\mathbf{96.11}_{\pm0.18}$ | $\mathbf{96.53}_{\pm0.12}$ | $96.12_{\pm0.12}$ |

Due to the space constraint on the main paper, we provide the experimental results on CIFAR-10 in Table 6. It can be seen that our proposed method achieves superior performance across various selection ratios on CIFAR-10. Since CIFAR-10 is relatively simple to classify for ResNet-50 models, various methods show only marginal accuracy differences, particularly at higher selection ratios. However, our method consistently outperforms existing baselines by a relatively larger margin. This highlights the effectiveness of our multimodal data selection strategy in improving model performance.

# E    GENERALIZATION ON VISION TRANSFORMER

To further demonstrate the generalization performance of our method on various architectures, we employ the Vision Transformer. The implementation is based on the public Github repository . Specifically, we utilize the ViT-small to train on the selected datasets. The experimental results are presented in Table 7. It can be observed that our method achieves the best performance compared to other baselines using ViT, validating the superior generalization performance on ViT architectures. Combined with the results on VGG-16 and DenseNet-121 in Section 4.2, we demonstrate that our selected datasets obtain a wide range of applications regardless of specific architectures.

---

https://github.com/kentaroy47/vision-transformers-cifar10

Table 7: The test accuracy (%) on CIFAR-10 with ViT-small.

| Method/Selection Ratio | 60% | 70% | 80% | 90% | 100% |
|---|---|---|---|---|---|
| Random | $78.98_{\pm0.28}$ | $80.30_{\pm0.36}$ | $81.33_{\pm0.10}$ | $82.63_{\pm0.18}$ | $84.00_{\pm0.32}$ |
| EL2N | $79.35_{\pm0.09}$ | $80.73_{\pm0.08}$ | $81.62_{\pm0.08}$ | $82.90_{\pm0.09}$ | $84.00_{\pm0.32}$ |
| MoSo | $79.45_{\pm0.11}$ | $80.27_{\pm0.23}$ | $81.82_{\pm0.15}$ | $82.92_{\pm0.34}$ | $84.00_{\pm0.32}$ |
| GraNd | $79.22_{\pm0.06}$ | $80.59_{\pm0.19}$ | $81.53_{\pm0.18}$ | $82.72_{\pm0.08}$ | $84.00_{\pm0.32}$ |
| Glister | $78.33_{\pm0.01}$ | $79.84_{\pm0.27}$ | $81.33_{\pm0.05}$ | $82.65_{\pm0.09}$ | $84.00_{\pm0.32}$ |
| Herding | $76.08_{\pm0.19}$ | $78.53_{\pm0.45}$ | $80.31_{\pm0.01}$ | $82.08_{\pm0.02}$ | $84.00_{\pm0.32}$ |
| CG-Score | $79.09_{\pm0.29}$ | $80.96_{\pm0.05}$ | $82.02_{\pm0.23}$ | $82.91_{\pm0.05}$ | $84.00_{\pm0.32}$ |
| Forgetting | $77.87_{\pm0.32}$ | $80.86_{\pm0.08}$ | $81.90_{\pm0.31}$ | $82.69_{\pm0.20}$ | $84.00_{\pm0.32}$ |
| Moderate-DS | $79.54_{\pm0.19}$ | $81.28_{\pm0.13}$ | $81.98_{\pm0.16}$ | $82.61_{\pm0.27}$ | $84.00_{\pm0.32}$ |
| Self-sup. prototypes | $79.24_{\pm0.16}$ | $80.34_{\pm0.21}$ | $81.66_{\pm0.25}$ | $82.86_{\pm0.19}$ | $84.00_{\pm0.32}$ |
| Ours | $\mathbf{80.21}_{\pm0.10}$ | $\mathbf{82.13}_{\pm0.23}$ | $\mathbf{82.32}_{\pm0.29}$ | $\mathbf{84.26}_{\pm0.05}$ | $84.00_{\pm0.32}$ |

# F   IMPLEMENTATION DETAILS OF TRAINING ViT-BASED MODELS ON IMAGENET-1K

We train ViT-Base and ViT-Large on the selected ImageNet-1k datasets based on the implementation of He et al. (2021), and we train Swin Transformer based on the implementation of Liu et al. (2021). Specifically, we fine-tune the pre-trained ViT-B and ViT-L with a batch size of 32, a base learning rate of $5e-4$, a layer decay of 0.65, and a weight decay of 0.05. We fine-tune the Swin Transformer with a weight decay of $1e-8$, a base learning rate of $2e-5$, a warmup learning rate of $2e-8$, and a batch size of 64.

# G   ILLUSTRATION OF IMAGE CORRUPTION TYPES

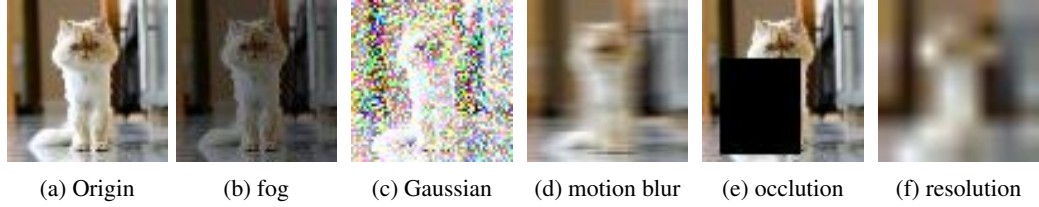

    (a) Origin      (b) fog      (c) Gaussian      (d) motion blur      (e) occlution      (f) resolution

Figure 8: Illustration of the image corruption types, including fog, Gaussian noise, motion blur, random occlution, and resolution.

# H   VISUALIZATION OF OUR METHOD IN NOISY CONDITIONS

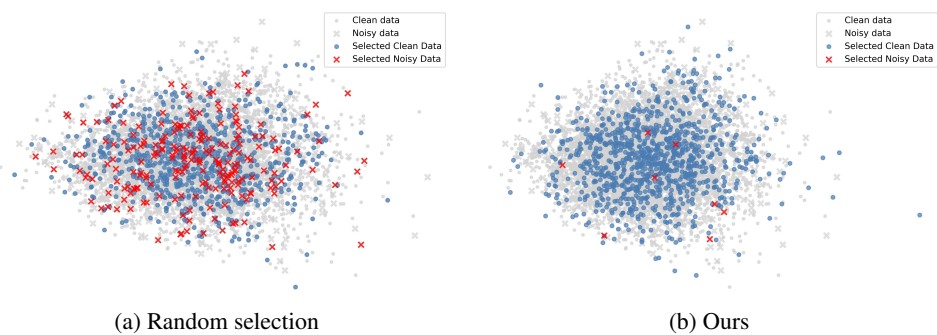

    (a) Random selection          (b) Ours

Figure 9: Illustration of the selected data in noisy conditions. The noisy ratio and selection ratio are 20%.

Table 8: Comparison of actual selection costs (h) of various methods.

| Method | Random | Herding | MoSo | Moderate | Glister | CG-Score | Forgetting | GraNd | EL2N | SSP | Ours |
|---|---|---|---|---|---|---|---|---|---|---|---|
| CIFAR-10 | 0 | 0 | 2.21 | 0.75 | 2.55 | 1.24 | 1.23 | 0.12 | 0.12 | 2.69 | 0.40 |
| Tiny-ImageNet | 0 | 0 | 3.17 | 1.50 | 5.75 | 0.15 | 2.16 | 0.21 | 0.21 | 5.14 | 0.76 |

## I    COMPARISON OF ACTUAL SELECTION COSTS

In Section 4.2, we present the comparison of tradeoffs in effectiveness and efficiency. In this section, to better understand the actual high efficiency of our method, we further present a direct comparison of the actual selection costs of different selection methods. The devices used are 4 NVIDIA RTX2080TI GPUs and an Inter(R) CPU E5-2678 @ 2.50GHz. We report the average costs across five independent runs and various selection ratios in Table 8, consisting of the costs of both fine-tuning the adapter and selection optimization. Consistent with the complexity analysis in Section 3, our approach achieves a superior efficiency compared to other baselines.

## J    TRADEOFF BETWEEN THE SELECTION COSTS AND ACCURACY

In this section, we present the tradeoff between accuracy and training costs across various selection ratios on CIFAR-100. The devices used are 2 NVIDIA RTX2080TI GPUs and an Intel(R) CPU E5-2678 @ 2.50GHz. The results are presented in Figure 10, where the baseline model uses the original dataset for training. It can be seen that with the increase in the selection ratios, the training costs approximate that of the baseline model. Notably, when the selection ratios are above 80%, the selected datasets can achieve lossless accuracy with lower training costs.

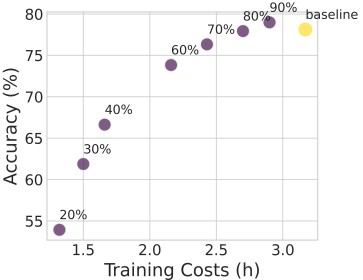

Figure 10: Relationship between accuracy and actual training costs.

It is important to emphasize that the practical gains from this approach are particularly significant in scenarios requiring the training of numerous models, such as neural architecture search (NAS) Ren et al. (2021); Zoph et al. (2018). In such applications, the reduction in training time and computational resources can result in substantial efficiency improvements, further amplifying the impact of our data selection framework.

## K    MORE ANALYSIS ON THE THRESHOLD $\theta$

Table 9: Effect of the threshold $\theta$ on Tiny-ImageNet.

| $\theta$ | 60% | 70% | 80% | 90% | 100% |
|---|---|---|---|---|---|
| $5 \times 10^{-4}$ | 44.29% | 46.02% | 47.41% | 49.30% | 49.36% |
| $5 \times 10^{-5}$ | 44.25% | 45.98% | 47.86% | 49.17% | 49.36% |

Threshold $\theta$ adjusts the gap between the actual and expected selection ratios. Employing a smaller $\theta$ makes weights $\boldsymbol{d}$ closer to the expected selection ratios while may increase the training costs for convergence. In our main results, we typically set $\theta$ as $5 \times 10^{-4}$. In this section, we employ a smaller threshold on Tiny-ImageNet, i.e., $5 \times 10^{-5}$, which indicates that the difference in the final sample numbers is less than 5. In Table 9, we select 60%-90% of the data from Tiny-ImageNet. It can be seen that the performance is robust to the change in $\theta$. With a tighter bound, the performance exhibits minimal differences.

Table 10: Comparison of noise reduction performance using text features (Ours) vs. average image features under varying noise and selection ratios. Noise proportion means the introduced noise ratio in the selected datasets.

| | Noise Ratio (%) | 20 | | 50 | | 70 | |
|---|---|---|---|---|---|---|---|
| | Selection Ratio (%) | 20 | 30 | 20 | 30 | 20 | 30 |
| Avg. Image Feat. | Noise Proportion (%) | 16.39 | 25.35 | 20.00 | 29.95 | 20.22 | 30.16 |
| | Accuracy (%) | 28.42 | 38.35 | 16.56 | 23.19 | 11.18 | 14.61 |
| Ours | Noise Proportion (%) | **0.24** | **0.32** | **0.43** | **0.68** | **0.80** | **4.30** |
| | Accuracy (%) | **46.05** | **58.34** | **52.56** | **60.72** | **51.50** | **56.80** |

## L EFFECTIVENESS OF TEXT MODALITY

Based on single-modal features, existing methods Xia et al. (2023) use the average image features as the prototype and calculate the Euclidean distance between the embedded image and the corresponding prototype. This distance is used to filter noisy labels. To further evaluate the effectiveness of our multi-modal framework, we use the average image feature to replace the text feature in Eq. 1, which is then used for selection optimization in Eq. 6.

In Table 10, we evaluate the noise robustness across various noise and selection ratios. It can be seen that using average image features leads to higher noise ratios compared to our method. This validates the effectiveness of complementary textual information provided by text features, leading to superior denoising performance.

## M NUMERICAL ANALYSIS ON THE SELECTED DATASETS

Table 11: Actual selection ratios on both Tiny-ImageNet and CIFAR-100.

| Expected Ratios | 20% | 30% | 40% | 60% | 70% | 80% | 90% |
|---|---|---|---|---|---|---|---|
| Tiny-ImageNet | 20.03%↑ | 30.03%↑ | 39.98%↓ | 59.97%↓ | 70.01%↑ | 80.02%↑ | 89.99%↓ |
| CIFAR-100 | 19.95%↓ | 30.02%↑ | 40.00%− | 60.04%↑ | 70.03%↑ | 80.04%↑ | 90.04%↑ |

We present the numerical analysis of the actual selection ratios on Tiny-ImageNet in Table 11. It can be seen that the actual selection ratios may be slightly higher or lower than the expected values, with minimal deviations. These deviations are within the theoretical gap bounds. Notably, when the actual selection ratios are below the expected ones, our method **selects fewer samples compared to other baselines**. Despite this reduction in training data volume, our method still achieves the best performance, highlighting the effectiveness of the selection strategy. Consequently, the performance improvements achieved are not due to the slight increases (e.g., less than 0.4% on average) in sample numbers but rather due to the strategic selection of informative samples, demonstrating the efficacy of our selection algorithm in optimizing the selected samples.

