# OpenReview forum: "A CLIP-Powered Framework for Robust and Generalizable Data Selection"
_ICLR.cc/2025/Conference — ICLR 2025 Spotlight_

### Official Review · Reviewer_vC5o · 2024-10-29

**Soundness:** 3
**Presentation:** 3
**Contribution:** 3
**Rating:** 6
**Confidence:** 3

**Summary:**

The paper proposes a CLIP-powered framework for data selection that addresses the limitations of traditional single-modality data selection methods by incorporating multimodal information. This framework utilizes a pretrained vision-language model (CLIP) to integrate image and text modalities, thereby enhancing data selection accuracy and robustness. The framework is built on three modules: Dataset Adaptation, Sample Scoring, and Selection Optimization. It leverages both semantic alignment and diversity scores to refine sample selection, removing redundant and noisy data to improve training efficiency and performance on benchmark datasets. Experiments demonstrate the method's superior performance in generalization and robustness compared to existing state-of-the-art approaches, especially in noisy environments.

**Strengths:**

1. Originality: The use of multimodal data via CLIP for data selection is innovative and distinguishes this work from single-modality approaches.
2. Quality: The methodology is backed by rigorous experiments, with a well-defined scoring system for sample selection (SAS and SDS).
3. Clarity: Visualizations effectively depict the advantages of multimodal selection over traditional methods.
4. Significance: This framework has broad applications, as it improves dataset quality and model robustness, potentially benefiting various domains using machine learning.

**Weaknesses:**

1. Complexity: While effective, the Selection Optimization module’s complexity might hinder its application to extremely large datasets without computational resources.
2. Generality: The framework is primarily tested on vision-based datasets; extending it to text or mixed-modality datasets may reveal further limitations. Future work could focus on enhancing the framework’s versatility across other modalities​

**Questions:**

1. Could the authors provide more detail on the computational efficiency of the Selection Optimization module for very large-scale datasets?
2. Has the framework been tested on datasets with high modality imbalance (e.g., more text than images), and how does it handle such scenarios?
3. For real-world noisy datasets, could the authors provide more information on how they define and quantify "noisy" samples?

---

> ### Author Response · Authors · 2024-11-20
> **Response to Reviewer vC5o (1/2)**
>
> Dear Reviewer vC5o:
>
> Thank you for providing insightful comment on our work. We appreciate your recognition of our work’s strengths and provide responses to address the comments as follows:
>
> - **Q1: Complexity: While effective, the Selection Optimization module’s complexity might hinder its application to extremely large datasets without computational resources.**
> - **A1:** Thank you for your thoughtful comment.
>
>     - Regarding computational complexity, as we have analyzed at the end of Section 3, this complexity is O(N), where N is the dataset size. The optimization module involves only numerical optimization without relying on deep models, ensuring high computational efficiency. For further details on computational costs, we kindly refer you to Table D-1 in Q3 and A3, where we provide the actual training costs for the selection optimization on large-scale ImageNet-1k.
>
>     -  Regarding parameter complexity, the module's parameter complexity is also O(N), which is negligible compared to deep model training. For instance, on ImageNet-1k with 1.2M samples, the parameter complexity of $d$ constitutes only **3.5%** of WRN-28-10 (36.5M), **5.5%** of ResNet-50 (23.5M), **1.5%** of ViT-B (86M), and **0.8%** of CLIP (ViT-B/32). Thus, the parameter complexity of the optimization module requires significantly less parameter complexity than these widely used models. This demonstrates that the optimization module requires significantly fewer parameters than these widely used models, ensuring its scalability to large-scale datasets with modest computational resources.
>
>   We hope this clarifies the efficiency of the selection optimization module and its feasibility for applications.
>
> - **Q2: Generality: The framework is primarily tested on vision-based datasets; extending it to text or mixed-modality datasets may reveal further limitations. Future work could focus on enhancing the framework’s versatility across other modalities.**
> - **A2:** Thank you for pointing out the generality of our framework and its potential for text or mixed-modality datasets. In our current work, following the task settings from [1-3], we focus on image datasets for data selection and enhance robustness and generalizability by introducing textual modality as complementary information, particularly for handling noisy or corrupted data.
>
>   We agree that extending the framework to other modalities, such as text-only or mixed-modality datasets, is a promising direction.  It is worth noting that research on data selection for text or mixed-modality datasets remains relatively limited compared to vision-based tasks, underscoring the potential impact of exploring these areas further. While such extensions may not be covered by this work, they represent an exciting avenue for future research, where enhancing the framework’s versatility across diverse modalities could unlock broader applications. We appreciate your suggestion and have incorporated this perspective into our discussion of future work.
>
> - **Q3: Could the authors provide more detail on the computational efficiency of the Selection Optimization module for very large-scale datasets?**
> - **A3:** Thanks for raising this question. As analyzed in Section 3 (lines 292-301), the computation complexity of the selection optimization module is O(N), where N is the dataset size. The optimization process relies solely on numerical optimization and does not involve deep architectures, making it inherently efficient and scalable for large-scale datasets.
>
>   To further address your concerns, we provide the actual training costs (h) for the selection optimization module, described in Section 3.3, on ImageNet-1k with a single V100 GPU across various selection ratios in the table below. The results, shown in the table below, demonstrate that the optimization process completes efficiently in just a few seconds, even for large-scale datasets.
>
> **Table D-1**: The actual training costs (h) for the selection optimization.
> |Selection Ratio (%)|90|80|70|60|40|30|20|
> |-|-|-|-|-|-|-|-|
> |Tiny-ImageNet|0.003|0.004|0.004|0.004|0.004|0.004|0.004|
> |ImageNet-1k|0.004|0.004|0.004|0.004|0.004|0.004|0.004|0.005|

---

> ### Author Response · Authors · 2024-11-20
> **Response to Reviewer vC5o (2/2)**
>
> - **Q4: Has the framework been tested on datasets with high modality imbalance (e.g., more text than images), and how does it handle such scenarios?**
> - **A4:** Insightful points. We would like to clarify that our work primarily focuses on image datasets, utilizing textual modalities to guide sample selection,  where the two modalities are set to be balanced. Our work is a pioneering effort in leveraging cross-modal information for image dataset selection. Compared with existing methods, our framework has the potential to be extended to the selection of multimodal data due to its multimodal nature. However, further optimization for such scenarios might be necessary, such as dynamic modality weight allocation mechanism and augmentation strategies, especially for modality-imbalanced datasets. We believe this will be a promising direction and will include relevant discussions in the paper.
>
> - **Q5: For real-world noisy datasets, could the authors provide more information on how they define and quantify "noisy" samples?**
> - **A5:** Good question. We follow previous works [1,4,5] to introduce symmetric noise into labels and corrutption into images as a way to simulate real-world noisy datasets. The noise is measured by the alignment between images and their associated labels. Results show that using multimodal semantic alignment can greatly improve robustness, as indicated in Table 2 and Figure 5 in the manuscript. However, we acknowledge that rigorously defining or quantifying noise in the real world is inherently challenging, as it often depends on the specific task, dataset, and noise characteristics. Although using multimodal semantics can alleviate the impact of noise, explicitly incorporating the characteristics of noise into the framework may lead to better results, we leave it to future exploration.
>
> [1] Moderate coreset: A universal method of data selection for real-world data-efficient deep learning, ICLR, 2023
>
> [2] Spanning Training Progress: Temporal Dual-Depth Scoring (TDDS) for Enhanced Dataset Pruning, CVPR, 2024
>
> [3] Data pruning via moving-one-sample-out. NeurIPS 2023
>
> [4] Confident learning: Estimating uncertainty in dataset labels. Journal of Artificial Intelligence Research. 2021 Apr 14;70:1373-411.
>
> [5] Combating noisy labels by agreement: A joint training method with co-regularization. CVPR 2020.

---

> ### Author Response · Authors · 2024-11-22
> **Looking forward to the reply**
>
> Dear reviewer vC5o:
>
> Thanks so much again for the time and effort in our work. According to the comments and concerns, we conduct the corresponding experiments and further discuss the related points. Besides, we have revised our paper and added a discussion to the main paper on page 10 in the revised version to further discuss the related points.
>
> As the discussion period is coming to a close, may I know if our rebuttal addresses the concerns? If there are further concerns or questions, please feel free to let us know. Thanks again for taking the time to review our work and provide insightful comments.

---

> ### Author Response · Authors · 2024-11-25
>
> Dear Reviewer vC5o,
>
> Considering the limited time available, and in order to save the reviewer's time, we summarized our responses here.
>
> Thank you for your constructive feedback and recognition of our work’s originality, quality, and significance. In response, we show additional experimental results and analysis:
>
> 1. We provide an analyses of both parameter and computational complexity of the selection optimization module, along with its computational efficiency on large-scale datasets (Table D-1). The results demonstrates that the optimization process completes efficiently within just a few seconds.
> 2. As suggested, we have included further discussion on high modality imbalance scenarios and outlined feasible directions for extending our method to such conditions. This has been added to the main paper on page 10 in the revised version.
>
> As the discussion period is about to close very soon, could we know if our responses addressed your concerns? Please feel free to let us know if there are any other concerns we can clarify.
>
> Thanks again for taking the time to review our work and provide insightful comments.

---

### Official Review · Reviewer_HyV4 · 2024-11-02

**Soundness:** 3
**Presentation:** 3
**Contribution:** 3
**Rating:** 8
**Confidence:** 3

**Summary:**

This paper proposes a novel CLIP-based data selection method leveraging multimodal information with an SGD optimization module. The experiments on several benchmarks show evident improvements over existing approaches.

**Strengths:**

- Connecting the text and image information for the dataset selection is novel. The proposed CLIP-based method implements the idea well.
- The SGD-based selection optimization with multi-objective is interesting. It is different from the existing sampling strategies with combinational optimization.
- Sufficient experimental results show the effectiveness, including different datasets, different settings, and different model architectures.

**Weaknesses:**

- The work relies on two adapters to project the CLIP features to the dataset-specific embedding space. The adapter training is based on perfect data. In fact, under the setting of noisy or corrupted data, this ideal data is inaccessible. Therefore, the experiments would be problematic. In Lines 514-515, the authors depict "is essential for effectively transferring the model’s generalization ability to target datasets". It seems the adapter has a large influence on the performance.  Will using the *imperfect* data to fine-tune the adapter degrade the method's effectiveness? This needs to be discussed.

- The class label in this work plays a role as a prototype, while previous work also uses prototypes. They [a] often use the average image features as the prototype and calculate the Euclidean distance between the embedded image and the corresponding prototype, similar to SAS (Eq. 1). Besides, they also use this distance to filter noisy labels [b]. Intuitively, we can use the average image feature to replace the text feature in Eq. 1 of the proposed method. Therefore, it is essential to discuss these results to convey to readers that introducing text is significant.

- MoSo is NeurIPS'23 rather than NeurIPS'24. Therefore, methods from 2024 have not been compared, e.g., [c]. Besides, some typical methods good at low sampling ratios are missing, e.g., [d].

- Typo: the symbol for the learnable parameter in Figure 2 should be $\mathbf{d}$ rather than $\mathbf{w}$.

[a] Moderate coreset: A universal method of data selection for real-world data-efficient deep learning, ICLR, 2023

[b] NGC: A unified framework for learning with open-world noisy data, ICCV, 2021

[c] Spanning Training Progress: Temporal Dual-Depth Scoring (TDDS) for Enhanced Dataset Pruning, CVPR, 2024

[d] Submodular Combinatorial Information Measures with Applications in Machine Learning, ALT, 2021

**Questions:**

- The text-image alignment is like using CLIP to measure the difficulty in learning the data. Noisy data or corrupted data are difficult to learn. However, will the selection also filter some important yet difficult data?

---

> ### Author Response · Authors · 2024-11-20
> **Response to Reviewer HyV4 (1/3)**
>
> Dear Reviewer HyV4:
>
> Thank you for providing meticulous review and insightful feedback on our work. We appreciate your recognition of our work’s strengths. For the comments and questions, we provide our response as follows.
>
> - **Q1: Will using the imperfect data to fine-tune the adapter degrade the method's effectiveness? This needs to be discussed.**
> - **A1:** Thank you for raising this insightful question. We would like to clarify that fine-tuning the adapter with imperfect data will **NOT** degrade the method's effectiveness. To validate this, we conducted a series of experiments:
>
> **1. Comparison with Clean vs. Imperfect Data:**
>
> We compared the performance of fine-tuning the adapter using clean data (**Ours**) and imperfect data (**Ours***) on CIFAR-100 and Tiny-ImageNet. As shown in Table C-1, the results demonstrate that using imperfect data yields comparable noise proportions (introduced into the selected datasets) and model performance. Additionally, as shown in Table C-2,  fine-tuning the adapter on corrupted data does not degrade the effectiveness, further showcasing the robustness of our method.
>
> **Table C-1**: Performance comparison between clean and data with noisy labels for fine-tuning the adapter on CIFAR-100 and Tiny-ImageNet with ResNet-50. Noise proportion means the noise ratio in the selected datasets.
> | Model    |   |CIFAR-100 |       |   Tiny-ImageNet   |       |
> |-|-|-|-|-|-|
> | | Selection Ratio (%) |  20   | 30      | 20       | 30      |
> | Ours | Noise Proportion (%)| 0.24| 0.25 | 0.28 | 0.27|
> |  | Acc. (%) | 45.63| 58.65 | 25.98| 32.21|
> | Ours*| Noise Proportion (%)| 0.25  | 0.32  | 0.26   | 0.23   |
> |  | Acc. (%)  | 46.05    | 58.34   | 26.09   | 33.13  |
>
> **Table C-2**: Performance comparison between clean and corrupted data for fine-tuning the adapter using Tiny-ImageNet with ResNet-50 and a 20% corruption ratio.
> | Selection Ratio (%) | 20 | 30|40|60|80|
> |-|-|-|-|-|-|
> |Ours|26.05|32.13|37.66|44.05|47.30|
> |Ours*|26.02|32.16|37.52|43.99|47.52|
>
>
>  **2. High-Noise Scenarios:**
>
> To further evaluate the robustness, we increased the noise ratio to as high as 70% and fine-tuned the adapter using imperfect data. Meanwhile, we also tested performance without using the adapter (i.e., leveraging CLIP's zero-shot capability). As shown in Table C-3, the results demonstrate that even under high-noise conditions, our method maintains low introduced noise in the selected datasets and achieves robust accuracy, validating the adapter's effectiveness.
>
> **Table C-3**:  Performance comparison under high-noise conditions with CIFAR-100 using different settings.
> |     | Noise Ratio (%)     | 20       |         | 50       |         | 70       |         |
> |-|-|-|-|-|-|-|-|
> || Selection Ratio (%) |20       | 30      | 20       | 30      | 20       | 30      |
> | Random             | Noise Proportion (%)| 20.80   | 19.83  | 20.32   | 30.10  | 20.83   | 29.93  |
> |                    | Acc. (%)           | 34.47    | 43.26   | 18.70    | 22.79   | 11.56    | 13.38   |
> | Ours w/o adapter   | Noise Proportion (%)| 0.33    | 0.52   | 1.37  | 0.74 | 1.70 | 6.42 |
> |                    | Acc. (%)           | 45.37    | 55.82   | 46.08    | 58.68   | 46.54    | 53.05   |
> | Ours*    | Noise Proportion (%)| **0.25** | **0.32** | **0.43** | **0.68**  | **0.80** | **4.30** |
> |                    | Acc. (%)           | **46.05** | **58.34**   | **52.56**    | **60.72**   | **51.50**    | **56.80**   |
>
> **Analysis:**
>
> CLIP's strong alignment capabilities, derived from extensive pretraining, make it inherently robust to noise. The adapter, designed for domain-specific transfer, is lightweight, with significantly fewer parameters (a simple linear layer constituting only 0.04% of CLIP ViT-B/32’s parameters) and minimal training iterations. This ensures the adapter complements rather than overshadows CLIP’s alignment capabilities. Our analysis further reveals that, across different noisy conditions, the alignment discrepancy between adapters trained on clean and noisy data is negligible (**<= 0.02%**), validating its robustness in noisy conditions. This leads to minimal impact on the SAS in Eq. 1 and the subsequent optimization module, ensuring that our method remains robust and effective, even when fine-tuned on imperfect data. We appreciate your attention to this critical aspect.

---

> ### Author Response · Authors · 2024-11-20
> **Response to Reviewer HyV4 (2/3)**
>
> - **Q2: Intuitively, we can use the average image feature to replace the text feature in Eq. 1 of the proposed method. Therefore, it is essential to discuss these results to convey to readers that introducing text is significant.**
> - **A2:** Thank you for the insightful comment. To address your suggestion, we conducted additional experiments to evaluate the performance of using average image features as prototypes, replacing the text features in Eq.1. The results, shown in Table C-4 and C-5, show that using average image features results in higher noise ratios in the selected datasets and notably lower accuracy compared to our method.
>   These findings validate that text features provide complementary semantic information, which enhances both noise robustness and accuracy. We have included these results in our ablation study in Section 4.5.
>
> **Table C-4**: Performance comparison of using text features (Ours) vs. average image features under varying noise and selection ratios with CIFAR-100. Noise proportion means noise ratio in the selected datasets.
> | Noise Ratio (%)|   | 20       |         | 50       |         | 70       |         |
> |-|-----|----------|---------|----------|---------|----------|---------|
> ||Selection Ratio (%)|20       | 30      | 20       | 30      | 20       | 30     |
> |Mean image feat.| Noise Proportion (%)| 16.39 | 25.35 | 20.00| 29.95| 20.22| 30.16|
> ||Acc. (%)|28.42|38.35|16.56|23.19|11.18|14.61|
> |Ours*|Noise Proportion (%) |**0.24** | **0.32** | **0.43**| **0.68** | **0.80** | **4.30** |
> ||Acc. (%)|**46.05**|**58.34**|**52.56**|**60.72**|**51.50**|**56.80**|
>
> **Table C-5**: Performance comparison of using text features (Ours) vs. average image features with a 20% corruption ratio on Tiny-ImageNet.
> | Selection Ratio (%)    | 20       |    30     | 40       |  60       | 80 |
> |-|-|-|-|-|-|
> |Avg. Image Feat.|12.74 |21.03|26.89|37.52|36.89|
> |Ours*|**26.02**|**32.16**|**37.52**|**43.99**|**47.52**|

---

> ### Author Response · Authors · 2024-11-20
> **Response to Reviewer HyV4 (3/3)**
>
> - **Q3: Methods from 2024 have not been compared, e.g., [c]. Besides, some typical methods good at low sampling ratios are missing, e.g., [d].**
> - **A3:** Thanks for suggesting additional references and comparisons.
>
> **1.** As suggested, we compared our method with the suggested work [c] on benchmark, noisy, and corrupted datasets (Table C-6/8/9), as well as the training efficiency (Table C-7). For CIFAR-10/100, we utilized the reported results from [c] and followed their implementation details to train the same models using our selected datasets.
>
>   The results show that while our method is slightly lower than TDDS at low selection ratios, ours can achieve higher accuracy at high selection ratios and noisy/corrupted datasets. Notably, our method demonstrates significant computational efficiency. As presented in Table C-7, it achieves a **10x** reduction in computational overhead compared to TDDS when calculating the average cost of obtaining one selected dataset or $k$ selected datasets.
>   Moreover, our method demonstrates significant advantages in other critical aspects. First, as shown in Tables C-8 and C-9, under noisy scenarios using CIFAR-100 and Tiny-ImageNet, our method exhibits superior noise robustness and accuracy compared to TDDS. Similarly, when evaluated on corrupted Tiny-ImageNet (20% corruption ratio) across various selection ratios, our method achieves superior robustness to data corruption.
>
> **Table C-6**: Comparison with [c] on CIFAR10/100 with ResNet-50. The accuracy of TDDS is obtained from its reported results.
> ||Selection Ratio (%)|30|50|70|
> |-|-|-|-|-|
> |CIFAR-10|TDDS|**93.92**|95.66|95.50|
> ||Ours|91.95|**95.74**|**95.86**|
> |CIFAR-100|TDDS|**66.56**|76.24|79.53|
> ||Ours|63.79|**77.15**|**79.93**|
>
> **Table C-7**: Training efficiency comparison between TDDS and Ours. Preparetion costs is the costs required prior to the selection begins. k is the number of selected datasets.
> ||TDDS|Ours|
> |-|-|-|
> |Preparation Costs (h)|4.15|**0.4**|
> |Selection Costs (h) | ~0|<0.005|
> |Total (k=1) (h)|4.15|**0.40**|
> |Total (k=5) (h)|0.83|**0.085**|
>
> **Table C-8**: Comparison with [c] under noisy conditions.
> |||CIFAR-100||Tiny-ImageNet||
> |-|-|-|-|-|-|
> ||Selection Ratio (%)|20|30|20|30|
> |TDDS|Noise Proportion (%)|18.54|28.93|17.75|29.59|
> ||Acc. (%)|35.15|45.66|20.67|25.52|
> |Ours|Noise Proportion (%)|**0.24**|**0.32**|**0.16**|**0.24**|
> ||Acc. (%)|**45.63**|**58.65**|**25.98**|**32.21**|
>
> **Table C-9**：Comparison with [c] on corrupted Tiny-ImageNet with a 20% corruption ratio.
> |Selection Ratio (%)|20|30|40|60|80|
> |-|-|-|-|-|-|
> |TDDS|23.67|29.24|35.13|41.04|44.22|
> |Ours|**26.05**|**32.13**|**37.66**|**44.05**|**47.30**|
>
>   **2.** Thank you for suggesting a comparison with [d]. We acknowledge the theoretical significance of this work, particularly its foundational contributions to establishing a strong foundation in connecting submodular functions and information-theoretic measures such as entropy and mutual information. However, as you noted, this work primarily emphasizes theoretical formulations and does not include empirical experiments, implementation details, or practical evaluations on data selection. As an alternative, in our work, we have compared several data selection methods inspired by their theoretical results, such as Glister and CG-Score, in Tab. 1-3 and Fig. 4-6.
>
> **3.** We have included the suggested works [c,d] in Section 2 of the revised manuscript. Specifically,
>
>   >-  Sec 2, paragraph 4, line 146: add reference [c] "such as temporal dual-depth scoring Zhang et al. (2024)"
>
>   >- Sec 2, paragraph 4, line 150: add reference [d] "and submodularity Iyer et al. (2021);"
>
> We hope these additions and clarifications address your concerns. Thank you again for your valuable suggestions.
>
> - **Q4: Typo: the symbol for the learnable parameter in Figure 2 should be d rather than w.**
> - **A4:** We appreciate your careful observation. We have corrected the symbol for the learnable parameter in Figure 2 in the revised version, replacing $w$ with $d$.
>
> - **Q5: The text-image alignment is like using CLIP to measure the difficulty in learning the data. Noisy data or corrupted data are difficult to learn. However, will the selection also filter some important yet difficult data?**
> - **A5:** Thanks for your insightful comments regarding the potential trade-off between filtering noisy data and retaining important but challenging data. We recognize that some complex data may be not selected, as distinguishing between noisy and genuinely difficult data is inherently challenging. However, we would like to clarify that our framework employs a multi-objective optimization strategy that balances semantic alignment (to reduce noise) and diversity to retain a wide variety of representative samples. This dual-focus approach minimizes the risk of discarding critical data while effectively reducing the impact of noisy data.

---

> ### Author Response · Authors · 2024-11-22
> **Looking forward to the reply**
>
> Dear reviewer HyV4:
>
> Thanks so much again for the time and effort in our work. According to the comments and concerns, we conduct the corresponding experiments and further discuss the related points. Additionally, according to your suggestions, we have revised the results of 4.2 and added additional ablation study results to 4.5 and Appendix J.
>
> As the discussion period is about to close, may I know if our rebuttal addresses the concerns? If there are further concerns or questions, we are willing to address them. Thanks again for taking the time to review our work and provide insightful comments.

---

> ### Comment · Reviewer_HyV4 · 2024-11-24
> **Experimental Setup and Explainations**
>
> Dear Authors,
>
> Thank you for sharing these additional experiments.
>
> Could you please elaborate further on the details of the experiments? Specifically:
>
> - Is the data used for training the adapter the same as that used for data selection and task model training?
> - What are the key differences between Table C-1 and Table C-2?
> - Why does using imperfect data for training degrade results in some cases while improving performance in others? What kind of data are selected or filtered with the imperfect fine-tuning?
> - In Table C-3, does `ours*` refer to fine-tuning the adapter with imperfect data or perfect data?
> - Additionally, it would be helpful if the final paper included examples of the corrupted data and showed visual results of different selection strategies.
>
> Thank you for your time and clarification.

---

> ### Author Response · Authors · 2024-11-25
>
> Dear Reviewer HyV4:
>
> Thank you for providing insightful questions and suggestions on our work.
>
> For the comments and questions, we provide more details of the experiments.
>
> - **Q6: Is the data used for training the adapter the same as that used for data selection and task model training?**
>
> - **A6:** Thanks for the question. Yes. The data used for training the adapter is the same as that used for selection and model training.
>
> - **Q7: What are the key differences between Table C-1 and Table C-2?**
>
> - **A7:** Thanks for the insightful comments. There are two types of noisy data in our main paper, noisy labels and corrupted images (which can be seen in Section 4.3 on pages 8 and 9).
>    - Table C-1 uses data with noisy labels,  where some sample labels are incorrectly flipped.
>    - Table C-2 uses data with corrupted images, where images are corrupted by Gaussian noise, random occlusion, resolution, fog, and motion blur.
>
> - **Q8: Why does using imperfect data for training degrade results in some cases while improving performance in others? What kind of data are selected or filtered with the imperfect fine-tuning?**
>
> - **A8:** Thanks for the insightful question. First, we want to emphasize that our method consistently achieves excellent performance, regardless of whether perfect or imperfect data is used for fine-tuning, and significantly outperforms other compared methods (as shown in Table 2 and Fig. 5 in Section 4.3).
>
> The variation in performance when using imperfect data for fine-tuning can be attributed to stochasticity in the training process and selection optimization. Training on noisy data may introduce slight instability, as the noise affects the optimization dynamics. Despite these factors:
> 1. The observed performance degradation remains minimal, averaging just 0.13%.
> 2.  While minor variations are observed between Ours and Ours*, both consistently outperform other methods by a significant margin. These results underscore the robustness and effectiveness of our approach, even under challenging conditions.
>
> As we have shown in Table C-1 and Table C-3, when using the imperfectly finetuned adapter for selection optimization, the introduced noise proportion remains significantly low. This indicates that the selection process **prioritizes clean data while effectively filtering out most of the noisy samples**, ensuring the robustness and reliability of the selected dataset.
>
> Thank you again for your valuable comment and for providing us with the opportunity to clarify this aspect.
>
> - ***Q9: In Table C-3, does ours* refer to fine-tuning the adapter with imperfect data or perfect data?**
>
> - **A9:** Thanks for the question. Ours* refers to fine-tuning the adapter with imperfect data.
>
> - **Q10 : Additionally, it would be helpful if the final paper included examples of the corrupted data and showed visual results of different selection strategies.**
>
> - **A10 :**  Thanks for the suggestion. We have included examples of the corrupted data in Figure 8 in Appendix G on page 17  in the revised version. Moreover, according to the reviewer's suggestion, we have also included visual results of the selection effectiveness in Figure 9 in Appendix H on page 17 in the revised version.

---

> > ### Comment · Reviewer_HyV4 · 2024-11-25
> >
> > I appreciate the authors' efforts to respond to my questions and improve the manuscript. The responses addressed my most concerns; therefore, I will raise the rating.
> >
> > Here is a minor problem that needs to be fixed in the revision:
> > "Ours* in Table C-3 refers to fine-tuning the adapter with imperfect data" somewhat conflicting with Table C-4 and Table A-1, where `ours` perhaps means the updated results of the training adapter with imperfect data. I would like to suggest the authors unify the symbol for fine-tuning with/without imperfect data in the manuscript.

---

> ### Author Response · Authors · 2024-11-26
>
> Dear reviewer HyV4,
>
> We would like to express our sincere gratitude to reviewer HyV4 for acknowledging our work and providing constructive suggestions. We will unify symbols in Tables C-4, A-1, and in the revised manuscript. Thanks again for the time and effort in reviewing our work.

---

### Official Review · Reviewer_vU6u · 2024-11-03

**Soundness:** 4
**Presentation:** 4
**Contribution:** 4
**Rating:** 8
**Confidence:** 4

**Summary:**

This paper argues that current image data selection methods are limited because they are unimodal. It proposes a multimodal method that uses the category texts from pretrained CLIP to complement images for more robust and generalized data selection. The proposed framework consists of three modules (1) dataset adaptation that integrates image and text adapters to transfer prior knowledge to the target data; (2) sample scoring that calculates the semantic alignment and diversity scores based on the multimodal features, measuring the image-text alignment as well as the local pattern variability; (3) Selection optimization that uses the two scores to select semantically representative and diverse samples, and introduces selection optimization to efficiently identify the ideal data subsets given an expected selection ratio through a multi-objective optimization strategy.

Post discussion comments:
The authors have addressed my concerns. With the additional clarifications and material, the paper is in much better shape. I am keeping my score at 8, which correctly reflects the quality of this work.

**Strengths:**

The idea of exploiting multimodal features from the CLIP model is interesting and plausible.

The proposed method is simple, which is a strength in my opinion.

The method is also efficient and is able to control the alignment, diversity and selection ratio in a multiobjective optimization efficiently.

The paper is well written and easy to follow.

The results are good.

**Weaknesses:**

The proposed method relies on a pretrained CLIP and hence any biases in the CLIP model will propagate to the selected dataset.

The proposed method optimizes alignment and diversity but does it have any indirect effect on bias in the dataset?

Is it possible to control bias in the dataset or in the subsequent models that are trained on the selected dataset?

Does the STE cause convergence issues?

The variable d (sample wise parameter) can be easily confused with d (feature dimension). Consider changing one of them. Also, I don’t see d in Fig. 1. Is “w: N x 1” the sample wise parameter d? Or am I missing something?

The caption of Tab.3 needs to be corrected. You can also merge Tab.3 with Tab.2.

**Questions:**

See above.

---

> ### Author Response · Authors · 2024-11-20
> **Response to Reviewer vU6u**
>
> Dear Reviewer vU6u:
>
> We sincerely thank you for the careful review and insightful questions/comments. For the comments and questions, we provide the responses here:
>
> - **Q1: The proposed method relies on a pretrained CLIP and hence any biases in the CLIP model will propagate to the selected dataset.**
> - **A1:** Thank you for pointing out the potential issue of bias propagation from the pretrained CLIP model to the selected dataset. In our method, apart from leveraging CLIP's alignment capabilities, the multi-objective selection optimization also incorporates a diversity scoring mechanism (SDS) to ensure a more representative and diverse subset of samples. By considering both semantic alignment and diversity during sample selection, the bias caused by CLIP could be further alleviated. However, we note that other state-of-the-art works based on pretrained CLIP share the raised limitation. Since our work primarily focuses on data selection, fully mitigating potential bias in pretrained multimodal models is, in our opinion, beyond its scope but of interest for our future work. Some feasible solutions to reduce bias in the data selection process may include fine-tuning CLIP or adapters with bias-aware loss functions, adding bias-related evaluation metrics, and so on.
>
>    We have added this discussion to the main paper on page 10.
> - **Q2: The proposed method optimizes alignment and diversity but does it have any indirect effect on bias in the dataset?**
> - **A2:** Thanks for the question. We acknowledge that investigating and addressing potential biases in training datasets is inherently challenging due to the complex nature of defining and quantifying bias in training datasets. In practice, training datasets are typically evaluated based on the model generalization performance trained using them.
>   Thus, in our work, we focus on validating the performance of the selected datasets across diverse architectures and scenarios. Specifically:
>   1. General performance validation: Performance across multiple deep architectures and datasets, as demonstrated in Fig. 4 and Tab. 1.
>   2. Robustness in Noisy Scenarios: Validation in noisy environments, shown in Tab. 2 and 3, and Fig. 5.
>   3. Generalization on Unseen Architectures and Benchmarks: Demonstrated in Tab. 4 and 5.
>
>   These evaluations collectively highlight the robustness and effectiveness of our method in improving performance across diverse conditions while indirectly addressing concerns about potential dataset bias. Thank you for bringing up this important point.
>
> - **Q3: Is it possible to control bias in the dataset or in the subsequent models that are trained on the selected dataset?**
> - **A3:** We appreciate your insightful comment. As demonstrated in Tables 2/3 and Figure 5, our method effectively controls noise-related bias in the selected datasets, achieving a significantly low noise ratio and reducing the risk of bias in subsequent models.
>   While our approach could suppress noise-related bias, controlling other forms of bias in datasets or downstream models depends on additional factors, such as inherent biases in the original dataset and the specifics of the downstream task. To control bias in the selected datasets, potential strategies may be feasible through bias-aware sample weighting and fine-tuning, which remains an important direction for future research. Thank you again for highlighting this critical aspect.
>
> - **Q4: Does the STE cause convergence issues?**
> - **A4:** Thanks for the question. We would like to clarify that STE will **NOT** cause convergence issues in our framework. To further address this concern, we provide the actual time costs (h) required for convergence using STE during the selection optimization process, as described in Section 3.3, on large-scale Tiny-ImageNet and ImageNet-1k across selection ratios on a single V100 GPU. As shown in the table below,  the convergence completes efficiently in just a few seconds.
>
> **Table B-1**: Convergence costs (h) of the selection optimization module.
> |Selection Ratio (%)|90|80|70|60|40|30|20|
> |-|-|-|-|-|-|-|-|
> |Tiny-ImageNet|0.003|0.004|0.004|0.004|0.004|0.004|0.004|
> |ImageNet-1k|0.004|0.004|0.004|0.004|0.004|0.004|0.004|0.005|
>
> - **Q5: The variable d (sample wise parameter) can be easily confused with d (feature dimension). Consider changing one of them. Also, I don’t see d in Fig. 1. Is “w: N x 1” the sample wise parameter d? Or am I missing something?**
> - **A5:** Thank you for pointing this out.
>   - We have renamed the variable representing the feature dimension from $d$ to $f_d$ throughout the manuscript to avoid ambiguity.
>   - Additionally, after reviewing, $w$ in Fig. 2 is the sample-wise parameter. We have fixed this point in Fig. 2 in the revised version.
> - **Q6: The caption of Tab.3 needs to be corrected. You can also merge Tab.3 with Tab.2.**
> - **A6:**  Thank you for the suggestion. We have corrected the caption of Tab.3 and merged it with Tab.2 in the revised version.

---

> > ### Comment · Reviewer_vU6u · 2024-11-22
> > **Final recommendation**
> >
> > Thank you for the detailed response to my comments. My concerns are well addressed. Please do incorporate the clarifications and additional material in the final paper.
> >
> > I am keeping my score the same.

---

> ### Author Response · Authors · 2024-11-22
> **Looking forward to the reply**
>
> Dear reviewer vU6u:
>
> Thanks so much again for the time and effort in our work. According to the comments and concerns, we conduct the corresponding experiments and further discuss the related points. Additionally, we have revised our paper for presentation clarity and added a discussion to the main paper on page 10 in the revised version to further discuss the raised points.
>
> As the discussion period is coming to a close, may I know if our rebuttal addresses the concerns? If there are further concerns or questions, please feel free to let us know. Thanks again for taking the time to review our work and provide insightful comments.

---

> ### Author Response · Authors · 2024-11-22
>
> Dear reviewer vU6u,
>
> We would like to express our sincere gratitude to reviewer vU6u for acknowledging our work and providing constructive suggestions. We will incorporate the clarifications and additional material in the final paper. Thanks again for the time and effort in reviewing our work.

---

### Official Review · Reviewer_QkGn · 2024-11-03

**Soundness:** 3
**Presentation:** 3
**Contribution:** 3
**Rating:** 8
**Confidence:** 4

**Summary:**

This paper introduces a CLIP-based multimodal data selection framework that enhances the robustness and generalizability of data selection by leveraging both image and text information. Firstly, the paper trains an adapter to transfer pretrained knowledge to the target data. Then, the similarity between the image and text is used to calculate the sampling score for subset selection. The authors evaluate their results in various popularly datasets, obtaining state-of-the-art results in almost all of them.

**Strengths:**

1. The method performs more robustly compared to other baseline data selection methods in various popularly datasets.
2. The algorithm is easy to understand and to follow.
3. The writing of this paper is overall great.

**Weaknesses:**

Some details and motivation should be explained further.
1. The method requires training an adapter with data from the training dataset for data selection. If the noise ratio is high (such as 50%), wouldn't it be a better option not to use an adapter?
2. Does the Actual Selection Costs in Appendix G include the training time for the Adapter? Is the comparison fair?
3. From the ablation experiment, it can be seen that selection loss has a significant impact on the final result. So, what is the sensitivity of this parameter?

**Questions:**

The authors should take the above questions into consideration: the utilization of the adapter, the comparison, and the parameter.
1. The method requires training an adapter with data from the training dataset for data selection. If the noise ratio is high (such as 50%), wouldn't it be a better option not to use an adapter?
2. Does the Actual Selection Costs in Appendix G include the training time for the Adapter? Is the comparison fair?
3. From the ablation experiment, it can be seen that selection loss has a significant impact on the final result. So, what is the sensitivity of this parameter?

---

> ### Author Response · Authors · 2024-11-20
> **Response to Reviewer QkGn**
>
> Dear Reviewer QkGn,
>
> We sincerely thank you for the careful review and insightful comments/questions. We appreciate your recognition of our work’s strengths and provide responses to address the comments raised.
>
> - **Q1: If the noise ratio is high (such as 50%), wouldn't it be a better option not to use an adapter?**
> - **A1:** Thanks for raising your insightful concerns. To further demonstrate the robustness of the adapters used in our method, we further perform a detailed evaluation under varying noise ratios (20%, 50%, and 70%) across several scenarios:
>    1) A random baseline;
>    2) Selection without using the adapter;
>    3) Selection using the adapter trained on the corresponding noisy datasets.
>
> We present the results in the table below. It shows that, under our framework, the proportion of noisy samples in the selected datasets remains considerably low across various noise ratios, regardless of whether the adapter is used or not. Although the difference in noise proportion between the settings with and without using the adapter is marginal, our method achieves higher accuracy using the adapter. This highlights that even under high-noise conditions, our approach can still ensure a strategical selection ability while maintaining denoising effectiveness, leading to better performance.
>
> **Table A-1**: Comparison of noise proportion and accuracy (%) with CIFAR-100 in high-noise conditions with and without using the adapter. Noise proportion means the introduced noise ratio in the selected datasets.
> | | Noise Ratio (%)  | 20| | 50  |  | 70| |
> |-|-|-|-|-|-|-|-|
> || Selection Ratio (%) |20       | 30      | 20       | 30      | 20       | 30      |
> | Random             | Noise Proportion (%)| 20.80   | 19.83  | 20.32  | 30.10  | 20.83| 29.93|
> |     | Acc. (%)           | 34.47    | 43.26   | 18.70    | 22.79   | 11.56    | 13.38   |
> | Ours w/o adapter   | Noise Proportion (%)| 0.33  | 0.52   | 1.37 | 0.74  | 1.70 | 6.42  |
> |    | Acc. (%)           | 45.37    | 55.82   | 46.08    | 58.68   | 46.54    | 53.05   |
> | Ours*               | Noise Proportion (%)| **0.24** | **0.32**  | **0.43** | **0.68**  | **0.80** | **4.30** |
> |   | Acc. (%)           | **46.05**    | **58.34**   | **52.56**    | **60.72**   | **51.50**    | **56.80**   |
>
> **Analysis:**
>
> CLIP's strong alignment capabilities, derived from extensive pretraining, make it inherently robust to noise. The adapter, designed for domain-specific transfer, is lightweight, with significantly fewer parameters (a simple linear layer constituting only 0.04% of CLIP ViT-B/32’s parameters) and minimal training iterations. This ensures the adapter complements rather than overshadows CLIP’s alignment capabilities. Our analysis further reveals that, across different noisy conditions (even high-noise conditions), the alignment discrepancy between adapters trained on clean and noisy data is negligible (<= 0.02%), validating its robustness in noisy conditions. This leads to minimal impact on the SAS in Eq. 1 and the subsequent optimization module, ensuring our method remains robust and effective, even when fine-tuned on imperfect data. We appreciate your attention to this critical aspect.
>
> - **Q2: Does the Actual Selection Costs in Appendix G include the training time for the Adapter? Is the comparison fair?**
> - **A2**: The selection costs reported in Appendix G **DO** include the training time for the adapter. Due to the lightweight architecture of the adapter (a linear layer) and the fast fine-tuning process, this training cost is minimal. We have highlighted it in the revised version in Appendix G. As all training costs are covered, the comparison is fair.
> - **Q3: From the ablation experiment, it can be seen that selection loss has a significant impact on the final result. So, what is the sensitivity of this parameter?**
> - **A3:** Thanks for the question. To further investigate the stability of the parameter in the selection loss, namely β in Eq.(6). We present additional experiment results on CIFAR-100 using ResNet-50 across various β values. In the table below, the results show that the accuracy is not sensitive to different β values, validating the robustness of the parameter.
>
> **Table A-2**: Stability analysis of the parameter for the selection loss with CIFAR-100 using ResNet-50.
> | β         | 1.5          | 2            | 3            | 5            | 7            |
> |-|-|-|-|-|-|
> | Accuracy (%)   | 78.90±0.05   | 78.98±0.09   | 78.89±0.03   | 78.87±0.02   | 78.90±0.06   |

---

> ### Author Response · Authors · 2024-11-22
> **Looking forward to the reply**
>
> Dear reviewer QkGn:
>
> Thanks so much again for the time and effort in our work. According to the comments and concerns, we conduct the corresponding experiments and further discuss the related points. Besides, according to your comments, we have revised our description of experiment settings in Appendix G for clarification.
>
> As the discussion period is about to close, may I know if our rebuttal addresses the concerns? If there are further concerns or questions, we are willing to address them. Thanks again for taking the time to review our work and provide insightful comments.

---

> > ### Comment · Reviewer_QkGn · 2024-11-22
> > **Final score**
> >
> > After reading the response, the authors addressed my concerns, and I raised my rating.

---

> ### Author Response · Authors · 2024-11-22
>
> Dear reviewer QkGn,
>
> We would like to express our sincere gratitude to reviewer QkGn for acknowledging our work and providing insighful comments. Thanks again for the time and effort in reviewing our work.

---

### Author Response · Authors · 2024-11-20
**General Response**

# General Description:
Dear Area Chairs and Reviewers,

We sincerely thank you for the time and effort in reviewing our work. We greatly appreciate the constructive feedback and the recognition from all reviewers—QkGn (R1), vU6u (R2), HyV4 (R3), and vC5o (R4)—highlighting key strengths of our work, including (1) novelty (R1, R2, R3, R4), (2) overall presentation clarity (R1, R2, R4), (3) sufficient experiments results (R2, R3), and (4) significance (R4). Besides, the concerns are mainly concentrated on (1) more experiments and analysis (R1, R3, R4), and (2) specific refinements to improve clarity (R2, R3).

# Additional Experiments and Analyses:
In the responses, we show additional experimental results and analysis, including:
1. Comparison of using and without using the adapter in high-noise conditions (R1: Q1)(Table A-1)
2. Evaluation and analysis of the dataset adaptation and selection optimization costs (R2: Q4, R4: Q3 )(Table B-1, D-1)
3. Stability analysis of the parameter of selection loss (R1: Q3)(Table A-2)
4. The effectiveness of the adapter in noisy conditions (R3: Q1)(Table C-1/2/3)
5. Validation of the significance of introducing text modality (R3: Q2)(Table C-4/5)
6. Comparison with the suggested work (R3: Q3)(Table C-6/7/8/9)

Thank you again for your thoughtful feedback and for helping us refine our work further.

Sincerely,

Authors of Submission 3088

---

### Meta-Review · Area_Chair_BdJe · 2024-12-20

**Metareview:**

This work proposes a new framework for data selection to address the computational overhead and the impact of noisy data when training deep learning models. The key idea is to leverage multimodal information to better select important data and this is achieved by utilising the CLIP model. A set of scores is developed and a multi-objective optimisation is implemented to realise data selection. Experimental study demonstrates the efficacy of the proposed method. Reviewers are overall positive on this work, indicating the strengths on its robust performance, easy to understand, simple and efficient, and its originality and significance. Meanwhile, the reviewers raise the issues related to the clarification of some details, the impact of bias, convergence, the presence of imperfect data, comparison with the last methods, and the complexity and generality of the method. The authors provide a high quality rebuttal and effectively address most of the issues. All the final ratings are on the positive side. By checking the submission, the reviews, and the rebuttals, AC agrees with the reviewers on their observations. Therefore, this work is recommended for acceptance.

**Additional Comments On Reviewer Discussion:**

The reviewers raise the issues related to the clarification of some details, the impact of bias, convergence, the presence of imperfect data, comparison with the last methods, and the complexity and generality of the method. The authors provide a high quality rebuttal and effectively address most of the issues. All the final ratings are on the positive side. By checking the submission, the reviews, and the rebuttals, AC agrees with the reviewers on their observations.

---

### Decision · Program_Chairs · 2025-01-22

Accept (Spotlight)